# Single MVA-SARS-2-ST/N Vaccination Rapidly Protects K18-hACE2 Mice against a Lethal SARS-CoV-2 Challenge Infection

**DOI:** 10.3390/v16030417

**Published:** 2024-03-08

**Authors:** Sabrina Clever, Leonard Limpinsel, Christian Meyer zu Natrup, Lisa-Marie Schünemann, Georg Beythien, Malgorzata Rosiak, Kirsten Hülskötter, Katharina Manuela Gregor, Tamara Tuchel, Georgia Kalodimou, Astrid Freudenstein, Satendra Kumar, Wolfgang Baumgärtner, Gerd Sutter, Alina Tscherne, Asisa Volz

**Affiliations:** 1Institute of Virology, University of Veterinary Medicine Hannover, Buenteweg 17, 30559 Hanover, Germany; sabrina.clever@tiho-hannover.de (S.C.); christian.meyer.zu.natrup@tiho-hannover.de (C.M.z.N.); schuenemann.lm@web.de (L.-M.S.);; 2Division of Virology, Department of Veterinary Sciences, LMU Munich, 85764 Oberschleißheim, Germany; l.limpinsel@gmx.de (L.L.); georgia.kalodimou@viro.vetmed.uni-muenchen.de (G.K.); astrid.freudenstein@viro.vetmed.uni-muenchen.de (A.F.); satyendra.kumar@viro.vetmed.uni-muenchen.de (S.K.); gerd.sutter@lmu.de (G.S.); alina.tscherne@viro.vetmed.uni-muenchen.de (A.T.); 3Department of Pathology, University of Veterinary Medicine Hannover, Buenteweg 17, 30559 Hanover, Germany; georg.beythien@tiho-hannover.de (G.B.); malgorzata.rosiak@tiho-hannover.de (M.R.); kirsten.huelskoetter@tiho-hannover.de (K.H.); katharina.manuela.gregor@tiho-hannover.de (K.M.G.); wolfgang.baumgaertner@tiho-hannover.de (W.B.)

**Keywords:** SARS-CoV-2, poxvirus, multivalent vaccine, K18-hACE2 mice

## Abstract

The sudden emergence of SARS-CoV-2 demonstrates the need for new vaccines that rapidly protect in the case of an emergency. In this study, we developed a recombinant MVA vaccine co-expressing SARS-CoV-2 prefusion-stabilized spike protein (ST) and SARS-CoV-2 nucleoprotein (N, MVA-SARS-2-ST/N) as an approach to further improve vaccine-induced immunogenicity and efficacy. Single MVA-SARS-2-ST/N vaccination in K18-hACE2 mice induced robust protection against lethal respiratory SARS-CoV-2 challenge infection 28 days later. The protective outcome of MVA-SARS-2-ST/N vaccination correlated with the activation of SARS-CoV-2-neutralizing antibodies (nABs) and substantial amounts of SARS-CoV-2-specific T cells especially in the lung of MVA-SARS-2-ST/N-vaccinated mice. Emergency vaccination with MVA-SARS-2-ST/N just 2 days before lethal SARS-CoV-2 challenge infection resulted in a delayed onset of clinical disease outcome in these mice and increased titers of nAB or SARS-CoV-2-specific T cells in the spleen and lung. These data highlight the potential of a multivalent COVID-19 vaccine co-expressing S- and N-protein, which further contributes to the development of rapidly protective vaccination strategies against emerging pathogens.

## 1. Introduction

Emerging infectious diseases pose a constant threat to global public health. Such pathogens can arise suddenly and have the potential to cause severe and also lethal disease in humans with rapidly increasing infection rates [1]. Changing ecological and environmental conditions and the more pronounced interaction with wildlife in different areas of the world have a significant impact on the increasing incidences of new emerging pathogens [2]. Other long-standing diseases like the seasonal flu and re-emerging diseases such as Ebola continue to cause disease and death in humans [3]. In addition to hygiene measures and several therapeutic approaches, the most important countermeasure against new emerging pathogens is the rapid availability of safe and protective vaccines [4]. This has been impressively demonstrated by the COVID-19 pandemic, caused by severe acute respiratory syndrome coronavirus 2 (SARS-CoV-2), when the availability of new vaccine technology ready to be used in humans significantly spurred the authorization of several SARS-CoV-2 vaccine candidates very rapidly [5]. These new approaches were mainly based on mRNA technology, protein-based vaccines, and vector vaccine technology using adenovirus-based vaccines [6]. So far, most licensed COVID-19 vaccines target the SARS-CoV-2 spike protein (S-protein) as a vaccine antigen, since it represents the major target for neutralizing antibodies [7,8]. Several studies confirmed the rapid activation of SARS-CoV-2 specific immune responses in humans. Based on this promising experience, several other vaccine candidates have been developed. Another promising approach for COVID-19 vaccine development using viral vector technology is based on the Modified Vaccinia Virus Ankara (MVA). MVA is a highly attenuated and replication-deficient vaccinia virus that is licensed as a safe third-generation vaccine against smallpox and serves as a potent vaccine vector system for the development of new candidate vaccines. Its replication deficiency in mammalian cells guarantees an important safety profile and the large DNA genome offers the opportunity to develop multivalent vaccine approaches [9]. In previous preclinical and clinical studies, we demonstrated the safety, immunogenicity, and protective efficacy of an MVA-based candidate vaccine expressing a prefusion-stabilized version of a SARS-CoV-2-S antigen (MVA-SARS-2-ST) [10]. However, the efficacy of COVID-19 vaccines targeting the S-protein has been hampered with the rapid emergence of new variants of SARS-CoV-2. Here, mutations specifically affecting the SARS-CoV-2 S-protein resulted in reduced protective efficacy of vaccines due to the very low neutralizing capacity against these new viral variants [11,12]. As a result, the pandemic continued with the rapid and constant emergence of new variants, which also resulted in altered transmissibility and severity of disease after infection. Especially the Delta and Omicron Variant were characterized by significant mutational changes in the S1-protein, which resulted in the escape of the immune responses [13]. Rather than developing constant updates and adaptations of licensed vaccines to the current viral variants, a more suitable strategy is the development of new vaccination approaches generating a broader protection against SARS-CoV-2 by multivalent vaccines [14,15]. Such a multivalent vaccine approach in addition to the well-established S-protein also includes more conserved antigens within the candidate vaccines [16]. The use of more conserved vaccine antigens to induce a broader protective immune response has been already confirmed for the Influenza A virus (IAV). Here, the use of the highly conserved nucleoprotein (NP) of the Influenza virus showed protective efficacy against different IAV strains. Different candidate vaccines targeting the nucleoprotein alone or in combination with other conserved proteins of IAV demonstrated the activation of increased titers of broadly reactive immune responses against different IAV strains in preclinical as well as clinical studies [17,18,19,20,21]. In another approach, the combination of different nucleoproteins and hemagglutinin from different IAV strains expressed by MVA confirmed the activation of more cross-reactive immune responses as an approach to contribute to the development of a universal flu vaccine [21,22]. The approach of using multivalent vaccine candidates to induce a broader reactive immune response has also been confirmed for SARS-CoV-2. Here, vaccines targeting the SARS-CoV-2 N-protein in addition to the well-established S-protein were confirmed to induce a robust protection against the ancestral SARS-CoV-2 as well as several variants of concern (VOCs) in different animal models [23,24,25,26,27,28]. The SARS-CoV-2 N-protein (N-protein) is 49.5 kDa in size and comprises 419 amino acids. The conserved regions of the N-protein, N-terminal domain, and C-terminal domain show strong similarities to the N-protein structures of other coronaviruses [29]. During the SARS-CoV-2 replication cycle, the N-protein mediates the genome encapsidation for the incorporation of the viral genome into a particle. Furthermore, the N-protein has also been considered to be involved in the immune evasion strategy of SARS-CoV-2 escaping the host immune response. The high conservation of the N-protein for different coronaviruses as well as its essential role in viral replication makes the N-protein a suitable candidate antigen for vaccine development [29]. Unlike the S-protein, the N-protein is not expressed on the surface of SARS-CoV-2 virus particles. Due to this, N-specific candidate vaccines should not induce SARS-CoV-2-neutralizing antibodies, but specific T cell responses should be expected [30]. Indeed, robust titers of N-specific antibodies as well as sufficient levels of N-specific T cells were abundantly found in patients infected with SARS-CoV-2 [31,32]. This has also been confirmed in previous studies, when immunization with monovalent N-based vaccine candidates already showed the immunogenic and protective potential in preclinical studies [33,34]. The suitability of the N-protein has also been demonstrated using MVA as a vaccine vector platform. Here, prime-boost vaccination studies in different preclinical animal models confirmed the immunogenicity and protective capacity of multivalent MVA-based vaccines against SARS-CoV-2 [23,28]. Another important goal for the development of vaccines against new emerging pathogens with pandemic potential is to rapidly induce protective immunity. This is especially important in the case of an emergency, when there is the necessity to rapidly stop viral transmission. Here, the availability of vaccines that induce protection after a single vaccination or even shortly after vaccination is important in an immediate public health response [35]. Since these previous data confirmed improved immunogenicity and efficacy of MVA-based COVID-19 vaccines targeting the S- and the N-protein, we aimed to evaluate the ability of these vaccines to rapidly protect against SARS-CoV-2. For this, we used the lethal K18-hACE2 mouse model and tested the immunogenicity and protective capacity of single and emergency vaccination with an MVA-based candidate vaccine co-expressing the S- and N-protein. We confirmed that our MVA-SARS-2-ST/N candidate vaccine (MVA-ST/N) robustly protected the brain and lungs against SARS-CoV-2 challenge infection after single immunization in the lethal K18-hACE2 mouse model using a more conventional vaccination regimen of 28 days. Moreover, MVA-ST/N vaccination just 2 days before lethal SARS-CoV-2 challenge infection resulted in a delayed onset of the clinical disease outcome in these mice.

## 2. Materials and Methods

### 2.1. Cell Cultures

DF-1 cells (ATCC^®^ CRL-12203™) were maintained in a VP-SFM medium (Thermo Fisher Scientific, Planegg, Germany), 2% heat-inactivated fetal bovine serum (FBS) (Thermo Fisher Scientific, Planegg, Germany), and 2% L-glutamine (Thermo Fisher Scientific, Planegg, Germany). Primary chicken embryonic fibroblasts (CEFs) were prepared from 10- to 11-day-old chicken embryos (SPF eggs, VALO, Cuxhaven, Germany) using recombinant trypsin (Tryple TM, Thermo Fisher Scientific, Planegg, Germany) and maintained in Dulbecco’s Modified Eagle’s Medium (DMEM), 10% FBS, and 1% Minimal Essential Medium (MEM) non-essential amino acid solution (Sigma-Aldrich, Taufkirchen, Germany). Vero and Vero E6 cells (ATCC CCL-81) were maintained in DMEM high glucose, 5% FBS, and 1% MEM non-essential amino acid solution (Sigma-Aldrich, Taufkirchen, Germany). Human HaCat cells (CLS Cell Lines Service GmbH, Eppelheim, Germany) were maintained in DMEM high glucose, 10% FBS, 1% MEM non-essential amino acid solution. All cells were cultivated at 37 °C and 5% CO_2_.

### 2.2. Plasmid Construction

The coding sequence of the full-length SARS-CoV-2 N-protein (SARS-2-N) was modified in silico to remove runs of guanines and cytosines and termination signals of vaccinia-virus-specific early transcription. The modified SARS-2-N cDNA was generated by DNA synthesis (Eurofins, Ebersberg, Germany) and cloned into the MVA transfer plasmid pLW-73-SARS-2-N, allowing the insertion of the target gene between the open reading frames (ORFs) of the viral genes MVA069R and MVA070L [36].

### 2.3. Generation of Recombinant Viruses

The generation of recombinant MVA-SARS-2-ST (MVA-ST) was described previously [10]. Recombinant MVA viruses MVA-SARS-2-N (MVA-N) and MVA-SARS-2-ST/N (MVA-ST/N) were generated following established protocols as described in previous studies [37,38,39]. The 90–95% confluent CEF or DF-1 cells were grown in six-well tissue culture plates (Sarstedt, Nümbrecht, Germany), infected with non-recombinant MVA or recombinant MVA-ST at a multiplicity of infection (MOI) of 0.05 and transfected with plasmid pLW-73-SARS-2-N using an X-treme GENE HP DNA Transfection Reagent (Roche Diagnostics, Penzberg, Germany) according to the manufacturer’s instructions to obtain MVA-N and MVA-ST/N, respectively. Subsequently, cell cultures were collected and recombinant MVA viruses were clonally isolated by serial rounds of plaque purification on CEF cell monolayers, monitoring for the co-expression of the green fluorescent marker GFP. To obtain vaccine preparations, recombinant MVA-N and MVA-ST/N were amplified on CEF or DF-1 cell monolayers grown in T175 tissue culture flasks, purified by ultracentrifugation through 36% sucrose and reconstituted to high-titer stock preparations in Tris-buffered saline (TBS) (pH = 7.4). To determine viral titers, plaque-forming units (PFUs) were counted following established protocols [37].

### 2.4. In Vitro Characterization of Recombinant MVA-N and MVA-ST/N

Genetic identity of MVA vector viruses were confirmed by a polymerase chain reaction (PCR) using viral DNA. Replicative capacity of recombinant MVA viruses was determined in multi-step-growth experiments on CEF or DF-1 and HaCat cells grown in six-well tissue culture plates. MVA-N, MVA-ST/N, and non-recombinant MVA were inoculated at MOI 0.05, and cell cultures were collected 0, 4, 8, 24, 48, and 72 h post infection (hpi), and titrated on 90–95% confluent CEF cell monolayers to define infectiousness in cell lysates (in PFU/mL).

### 2.5. Western Blot Analysis of Recombinant Proteins

To confirm the unimpaired and stable expression of recombinant SARS-2-N and SARS-2-ST proteins, monolayers of 90–95% confluent DF-1 cells were infected at MOI of 5 with recombinant or non-recombinant MVA or remained un-infected (mock) using protocols as described previously [10,40]. Cell lysates were prepared at the indicated time points and stored at −80 °C. Proteins from lysates were separated by SDS-PAGE and transferred to a nitrocellulose membrane by electroblotting. The blots were blocked in a phosphate buffered saline buffer (PBS) containing 5% non-fat dried milk powder (PanReac AppliChem, Darmstadt, Germany) and 0.1% Tween20 (Sigma-Aldrich, Taufkirchen, Germany) and were incubated overnight with primary antibodies targeting the S2-domain of the S-protein (1A9, 1:4000; GeneTex, Irvine, CA, USA) or the N-protein (HL249, 1:4000; GeneTex, Irvine, CA, USA) of SARS-CoV-2. As a positive control, the blots were also incubated with a monoclonal antibody targeting Hsp90 (C45G5, 1:2000; Biozol, Eching, Germany). Subsequently, blots were washed twice with PBS/0.1% Tween20 and incubated with anti-mouse IgG-HRP (1:5000; Agilent Dako, Glostrup, Denmark) or anti-rabbit IgG-HRP (1:5000; Cell Signaling, Frankfurt am Main, Germany). Membranes were washed and developed by a SuperSignal^®^ West Dura Extended Duration substrate (Thermo Fisher Scientific, Planegg, Germany). Chemiluminescence was visualized using the ChemiDoc MP Imaging System (Bio-Rad, Munich, Germany).

### 2.6. Immunostaining of Recombinant SARS-2-ST and SARS-2-N Proteins

The detection of SARS-2-ST was performed using protocols as described before with minor changes [10,40]. Vero cells were infected at MOI of 0.1 with recombinant MVA-N and MVA-ST/N or non-recombinant MVA or remained un-infected and were incubated at 37 °C for 17 h. Cells were fixed with 4% paraformaldehyde (PFA) for 10 min on ice, washed twice with PBS, and permeabilized with PBS/0.1% Triton X-100 (Sigma-Aldrich, Taufkirchen, Germany). Permeabilized cells were probed with monoclonal antibodies directed against the S2-domain of SARS-2-S (1:1000; GeneTex, Irvine, CA, USA) or SARS-2-N (HL249; 1:1000; GeneTex, Irvine, CA, USA) for 1 h at room temperature. Polyclonal goat anti-mouse (1:1000; Life Technologies, Darmstadt, Germany) or polyclonal goat anti-rabbit (1:1000; Life Technologies, Darmstadt, Germany) secondary antibodies, diluted in PBS/0.5% BSA, were used to visualize S- and N-specific staining by red and green fluorescence, respectively. Nuclei were stained with 1 µg/mL of 4,6-diamidino-2-phenylindole (DAPI) (Sigma-Aldrich, Taufkirchen, Germany) and cells were visualized using the Keyence BZ-X700 microscope (Keyence, Neu-Isenburg, Germany) with a × 100 objective.

### 2.7. Viruses

SARS-CoV-2 (isolate Germany/BavPat1/2020, NR-52370) was obtained by BEI Resources, NIAID, NIH, and propagated in Vero cells cultured in DMEM (Sigma-Aldrich) supplemented with 2% FBS, 1% penicillin–streptomycin (P/S), and 1% L-glutamine at 37 °C. All experiments and applications with SARS-CoV-2 were performed in BSL-3 laboratories and BSL-3 stables at the Research Center for Emerging Infections and Zoonoses (RIZ), University of Veterinary Medicine Hannover, Germany.

### 2.8. Immunization and Infection of K18-hACE2 Mice

Immunizations were performed using intramuscular applications into the quadriceps muscle of the left hind leg with vaccine suspension containing 1 × 10^8^ PFU of MVA, MVA-ST, MVA-N, or MVA-ST/N in a volume of 50 µL. In total, 50 µL of PBS (mock) was administered as a control. For challenge infection studies with SARS-CoV-2, female and male K18-hACE2 mice (B6.Cg-Tg (K18-ACE2)2Prlmn/J) were kept in groups of two to five animals in individually ventilated cages (IVCs, Tecniplast). All performances working with infectious SARS-CoV-2 in the animal stables and laboratories were conducted in BSL-3 facilities at the Research Center for Emerging Infections and Zoonoses (RIZ), University of Veterinary Medicine, Hanover, Germany. A total of 28 or 2 days after vaccination, all mice were anesthetized followed by an intranasal infection with a 3.6 × 10^4^ TCID_50_ of SARS-CoV-2 (isolate Germany/BavPat1/2020, NR-52370) obtained from BEI Resources, NIAID, NIH. After SARS-CoV-2 infection, body weight of all mice was measured daily and signs for clinical disease were monitored at least twice a day using a clinical score sheet. Symptoms were assigned to the following categories: the cardiovascular system, fur/skin condition, respiratory tract, social behavior/general condition/locomotion, and neurological abnormalities. Respiratory signs were additionally divided into upper and lower respiratory tracts with focus on nasal discharge and tachypnoea, respectively. Clinical symptoms were assigned for clinical scores using the clinical score sheet and further combined as a total clinical score for each mouse. Exceeding weight loss from the initial body weight over 20% as well as a defined cumulative clinical score were declared as experimental endpoints. 

### 2.9. Measurement of Viral Burden

After the euthanization of the mice, the right lung lobe and brain tissue from the right brain hemisphere were sampled and homogenized in 1 mL of DMEM containing antibiotics (penicillin and streptomycin, Gibco). Homogenization was performed with the TissueLyser-II (Qiagen). Titers of infectious SARS-CoV-2 in the homogenates were determined by incubation on Vero cells and calculated as median TCID_50_ units. For this, Vero cells were cultured in DMEM containing 5% FBS and transferred onto 96-well plates. Samples of homogenized lungs and brains were added in serial 10-fold dilutions onto the cells and incubated for 96 h at 37 °C. After incubation, Vero cells were evaluated for cytopathic effects and the TCID_50_ unit per mL was calculated using the Reed–Muench method.

### 2.10. Quantitative Real-Time RT-PCR

Oropharyngeal swabs were taken at 4 days post infection (4 dpi) and on the day of death (final) and incubated in 1 × PBS containing antibiotics (penicillin and streptomycin, Gibco). After euthanization, the right lung lobe and brain tissue from the right brain hemisphere were sampled and homogenized within 1 mL of DMEM containing antibiotics (penicillin and streptomycin, Gibco). To determine viral RNA titers in lungs and brains of SARS-CoV-2-infected mice, isolated RNA from homogenates was amplified using RT-qPCR (quantitative real-time reverse transcription PCR) targeting the RNA-dependent RNA polymerase (RDRP) of SARS-CoV-2. The isolation of RNA was conducted with the KingFisher Flex and NucleoMag RNA kit following the manufacturer’s protocol. Isolated RNA was incubated with the Luna^®^ Universal One-Step RT-qPCR Kit (NEB #E3005, New England Biolabs GmbH, Frankfurt am Main, Germany) in a CFX96-Touch Real-Time PCR system (Bio-Rad) using the RT-qPCR program. The detection of RNA targeting RDRP of SARS-CoV-2 was conducted using the following primers: SARS-2-IP4, forward primer (5′-GGTAACTGGTATGATTTCG-3′), reverse primer (5′-CTGGTCAAGGTTAATATAGG-3′), and probe (5′-TCATACAAACCACGCCAGG-3′ [5′] FAM, [3′] BHQ-1). Incubation started with reverse transcription at 55 °C for 10 min followed by denaturation at 95 °C for 1 min. In 44 cycles, samples were heated at 95 °C for 20 sec for denaturation and at 56 °C for 30 s for annealing and elongation. Measurements of relative fluorescence units (RFUs) at the time point of the termination of the elongation step were conducted. A standard RNA transcript was used to correlate Ct values of the samples and calculate the quantity of viral copy numbers per µL of total RNA.

### 2.11. PRNT_50_

PRNT_50_ assays (plaque reduction neutralization tests) using sera of the mice were performed to determine the titers of neutralizing antibodies (nABs) against SARS-CoV-2 (isolate Germany/BavPat1/2020, NR-52370) obtained from BEI Resources, NIAID, NIH. Besides some modifications, the protocol was performed as described previously [41]. All serum samples were initially heat-inactivated at 56 °C for 30 min. Two-fold dilutions of all sera were conducted on 96-well plates in duplicates in a total volume of 50 μL of DMEM. Diluted sera were further mixed with 50 μL of a SARS-CoV-2 virus suspension (600 TCID_50_) per well and incubated at 37 °C for 1 h. After this, the mix was transferred onto Vero E6 cells (ATCC, CRL-1586), which were cultured in 96-well plates and incubated for 45 min. Meanwhile, Avicel RC-591 (Dupont, Nutrition & Biosciences, New York, NY, USA) and pre-warmed DMEM were mixed 1:1 and after incubation, 100 μL was added to the cell plates and further incubated for 24 h. The final fixation of the cells was conducted with 4% formaldehyde in PBS for further staining assays. Cells were stained targeting the SARS-CoV-2 N-protein with a polyclonal anti-rabbit antibody (Sino Biological, 40588-T62; 1:2000, Beijing, China). A secondary HRP-labeled goat anti-rabbit IgG (Agilent Dako, P044801-2; 1:1000, Santa Clara, CA, USA) was used to generate a signal by processing a precipitate-forming TMB substrate (True Blue, KPL SeraCare, 5510-0030, Milford, MA, USA). The amount of SARS-CoV-2-infected cells/wells was measured with an ImmunoSpot reader (CTL Europe GmbH). The serum neutralization titer (PRNT_50_) was calculated, which indicates the reciprocal of the highest serum dilution leading to the decrease of more than 50% of the plaque formations.

### 2.12. Antigen-Specific IgG ELISA

SARS-2-N-specific serum IgG titers were measured by ELISA as described previously with minor modifications [10,40]. ELISA plates (Nunc MaxiSorp Plates, Thermo Fisher Scientific) were coated with 50 ng/well of recombinant SARS-CoV-2 N-protein (AcroBiosystems) overnight at 4 °C. Plates were washed with PBS/0.05% Tween20 (PBS-T) (Sigma-Aldrich, Taufkirchen, Germany) and blocked with a blocking buffer containing 1% BSA (Carl Roth GmbH + Co. KG, Karlsruhe, Germany) and 0.15 M sucrose (Sigma-Aldrich, Taufkirchen, Germany) dissolved in PBS. Plates were probed with sera serially diluted 3-fold in PBS/1% BSA (Carl Roth GmbH + Co. KG), starting with a dilution of 1:100. Plates were probed with goat anti-mouse IgG HRP (1:2000; Agilent Dako, Glostrup, Denmark). After washing, plates were incubated with a 3,3′,5,5′-Tetramethylbenzidine (TMB) Liquid Substrate System for ELISA (Sigma Aldrich, Taufkirchen, Germany) and the reaction was stopped using a Stop Reagent for the TMB substrate (450 nm, Sigma-Aldrich). The absorbance was measured at 450 nm with a 620 nm reference wavelength using a Spark^®^ multimode microplate reader (Tecan Trading AG, Männedorf, Switzerland). ELISA data were normalized using the positive control SARS-CoV-2 N-protein antibody (clone HL455-MS, 1:10.000; GeneTex, Irvine, CA, USA). The cut-off value for positivity was determined by calculating the mean OD 450 nm values of the mock vaccination/mock challenge control group plus 6 standard deviations (mean + 6 SD).

### 2.13. T Cell Analysis by Flow Cytometry

To characterize T cells in the spleen, lung, brain, and whole blood, cells were isolated and analyzed using flow cytometry. Lung and brain tissues were cut with scissors into small pieces prior to digestion with a Roswell Park Memorial Institute (RPMI) 1640 medium supplemented with 10% FCS, 0.5 mg/mL of DNase (lung) or 0.1 mg/mL of DNase (brain) (Roche, Merck KGaA, Darmstadt, Germany), and 1 mg/mL of collagenase (GENAXXON bioscience GmbH, Ulm) for 30 min at 37 °C. Digested lung and brain tissue as well as spleens were smashed through a 70 µm strainer (Falcon^®^, Sigma-Aldrich, Taufkirchen, Germany) and again flushed with RPMI-10 (RPMI 1640 medium containing 10% FBS, 1% penicillin–streptomycin; Sigma-Aldrich, Taufkirchen, Germany). Next, red blood cells were lysed while incubating the cell suspension with a Red Blood Cell Lysis Buffer (Sigma-Aldrich, Taufkirchen, Germany). After washing once, cells were resuspended in 1 mL of a RPMI-10 medium. Cell counts for the lung and spleen were measured with the MACSQuant (Miltenyi Biotec B.V. & Co. KG, Bergisch Gladbach, Germany) and dead cells were visualized by using propidium iodide (PI, 1:10). Approximately 3 × 10^5^ cells of isolated splenocytes and lung cells in 50 µL of PBS supplemented with 3% FCS, as well as 50 µL of whole blood and 50 µL of an isolated brain cell suspension, were first incubated with SARS-2-S-, SARS-2-N-, or MVA-peptide-loaded tetramers (Tetramer Shop). The H-2kb tetramers were loaded with the immunodominant S-peptide S_V8L_ (539–546, VNFNFNGL [42], PE-conjugated), the immunodominant N-protein peptide N_219_ (219–227, LALLLLDRL [43], Brillant Violet 421-conjugated), or the immunodominant peptide for the vaccinia virus WR epitope B8R_20–27_ (MVA_B8R_, 20–27, TSYKFESV [44], APC-conjugated) for the analysis of specific T cells. Additional surface marker staining was conducted using monoclonal antibodies against CD3-PE-Vio770 (Miltenyi Biotec), CD4-FITC (Miltenyi Biotec), CD8-PerCP (Miltenyi Biotec), and CD44-BV510 (BioLegend, San Diego, CA, USA) for the initial characterization of T cells. To exclude dead cells, a LIVE/DEAD™ Fixable Near-IR Dead Cell Stain Kit (Invitrogen™, Thermo Fisher Scientific, Waltham, MA, USA) was used according to the manufacturer’s instructions. Cells were measured with the MACSQuant Analyzer 10 (Miltenyi Biotec) and analyzed with MACSQuantify Software (Miltenyi Biotec, version number: 2.13.3). Samples from unvaccinated and unchallenged K18-hACE2 mice, as well as unstained samples from vaccinated and challenged mice, were used to subtract background signaling especially for determining a tetramer-specific signal. Using gating strategies, specific CD8^+^ T cells (indicated by anti-CD3, anti-CD8, and tetramer staining) were also characterized as activated T cells (CD44^+^).

### 2.14. T Cell Analysis by Enzyme-Linked Immunospot (ELISpot)

To determine specific T cells in the spleen and lung, cells were isolated and re-stimulated using single peptides and one overlapping peptide pool. For this, lung tissues were cut with scissors into small pieces prior to digestion with RPMI 1640 supplemented with 10% FCS, 0.5 mg/mL of DNase (Roche, Merck KGaA, Darmstadt, Germany), and 1 mg/mL of collagenase (GENAXXON bioscience GmbH, Ulm) for 30 min at 37 °C. Digested lung tissue as well as spleens were smashed through a 70 µm strainer (Falcon^®^, Sigma-Aldrich, Taufkirchen, Germany) and again flushed with RPMI-10 (RPMI 1640 medium containing 10% FBS, 1% penicillin–streptomycin; Sigma-Aldrich, Taufkirchen, Germany). Next, red blood cells were lysed while incubating the cell suspension with a Red Blood Cell Lysis Buffer (Sigma-Aldrich, Taufkirchen, Germany). After washing once, cells were resuspended in 1 mL of the RPMI-10 medium. Cell counts were measured with the MACSQuant (Miltenyi Biotec B.V. & Co. KG, Bergisch Gladbach, Germany) and dead cells were visualized by using propidium iodide (PI, 1:10). Approximately 3 × 10^4^ cells of isolated splenocytes and lung cells were stimulated with the immunodominant peptides S_V8L_ (539–546, VNFNFNGL) of the S-protein or N_219_ (219–227, LALLLLDRL) of the N-protein of SARS-CoV-2 or B8R_20–27_ of MVA (MVA_B8R_; 20–27, TSYKFESV). Additionally, a peptide pool comprising the whole N-protein of SARS-CoV-2 was used for stimulation, which consists of 59 overlapping peptides (1 µg of peptide/mL of RPMI 1640). Peptides and the peptide pool were obtained by JPT Peptide Technologies (Berlin, Germany). Non-stimulated cells, phorbol myristate acetate (PMA), and ionomycin (SIGMA-ALDRICH, Taufkirchen, Germany) were used as negative and positive controls of stimulation. Cells together with stimulants were placed on plates with PVDF membranes (Mabtech, Nacka, Sweden), which were pre-coated with a mouse-specific anti-IFN-γ monoclonal antibody (Mabtech, Nacka, Sweden) and incubated for 36 h at 37 °C. The IFN-y production of stimulated cells was determined by a biotinylated anti-IFN-γ monoclonal antibody. After incubation and several washing steps, streptavidin ALP followed by a BCIP/NBT-plus substrate was added. Spots were scanned and analyzed using ImmunoSpot by C.T.L. and counted with an automated ELISpot Reader ImmunoSpot S6 ULTIMATE UV Image Analyzer (Immunospot, Bonn, Germany) and further analyzed with ImmunoSpot 7.0.20.1 software. The background signal, determined using control samples without peptide stimulation, was subtracted from each sample value.

### 2.15. Immunohistochemistry of Lung and Brain Tissue

Immunohistology was performed on formalin-fixed, paraffin-embedded tissue sections as previously described to detect and/or quantify the presence of a SARS-CoV-2 antigen [45,46]. The Dako EnVision+ polymer system (#K4001 and #K4003, Dako Agilent Pathology Solutions, Santa Clara, CA) served for immunolabeling of the SARS-CoV-2 antigen using either a polyclonal rabbit antibody specific for the S2-subunit (SARS-CoV-2-S, #40590-T62, 1:4000, Sino Biological Europe GmbH, Eschborn, Germany) on lung tissue or a monoclonal mouse antibody directed against the SARS-CoV-2 N-protein (SARS-CoV-2-N, #40143-181 MM05; 1:16,000, Sino Biological Europe GmbH, Eschborn, Germany) on brain tissue. Heat-induced antigen retrieval was performed in Na2H2EDTA and citrate-Na2H2EDTA for 20 min in a microwave at 800 W, respectively. Primary antibodies were incubated overnight at 4 °C. Mock-infected K18-hACE2 mice served as negative controls. In addition, specific primary antibodies were substituted with serum from non-immunized rabbits or ascites from non-immunized BALB/cJ mice as additional negative controls.

### 2.16. Digital Image Analysis

Tissue sections immunostained for SARS-CoV-2-S and -N were digitalized using the Olympus VS200 slide scanner (Olympus Deutschland GmbH, Hamburg, Germany) and a digital image analysis was performed using the open source software package QuPath (version 0.4) for a digital pathology image analysis [47]. For each animal, the evaluation was performed on one digital slide containing whole tissue sections of the right cranial and medial lung lobes or one longitudinal brain section including the cerebrum and brain stem. For all analyzed images, an individual color deconvolution was performed for the optimal identification of the hematoxylin and DAB signal on the slides.

Lung tissue was detected automatically on the analyzed slides using a stain-specific pixel classifier based on digital thresholding of the green color channel and the size exclusion of objects smaller than 106 µm^2^. Individual cells were subsequently identified using an automated script of the QuPath feature “cell detection”. Within this feature, individual nuclei were identified based on a nuclear hematoxylin stain threshold and size exclusion of objects smaller than 10 µm^2^ or larger than 400 µm^2^. The expansion of the digital cell body was set to a radius of 10 µm around each detected nucleus. To determine the amount of immunolabeled cells, an object classifier based on a random tree machine learning algorithm was trained by a veterinary pathologist on three representative images of the SARS-CoV-2-S staining by annotating ≥ 100 immunopositive and -negative cells per image. The percentage of immunolabeled cells for each annotated tissue section was calculated automatically. 

On brain sections, tissue was detected using individually adjusted pixel classifiers, thresholding the average signal of the red, green, and blue channel. Cerebral and brain stem tissue was annotated by a veterinary pathologist as regions of interest (ROIs). Immunolabeled areas within these ROIs were thereafter assessed using a pixel classifier measurement based on one DAB channel threshold for all slides.

### 2.17. Statistical Analysis

Data were prepared using GraphPad Prism 9.0.0 and expressed as the mean ± standard error of the mean (SEM) or median ± interquartile range. Data were analyzed by a One-way ANOVA or Kruskal–Wallis Test to compare 3 or more groups. A *p* value of less than 0.05 was used as the threshold for statistical significance.

### 2.18. Study Approval

All animal experiments working with the infectious SARS-CoV-2 virus under BSL-3 conditions were handled in compliance with the European and national regulations for animal experimentation (European Directive 2010/63/EU; Animal Welfare Acts in Germany) and the “Niedersächsisches Landesamt für Verbraucherschutz und Lebensmittelsicherheit” (LAVES, Lower Saxony, Germany).

## 3. Results

### 3.1. Generation and Characterization of the Candidate Vaccines MVA-N and MVA-ST/N

For the generation of MVA-based candidate vaccines expressing SARS-2-N either alone or in combination with SARS-2-ST, cDNA containing the entire gene sequence encoding for SARS-2-N from the virus isolate Wuhan HU-1 (GenBank 126 accession no. MN908947.1) was placed under the transcriptional control of the enhanced synthetic vaccinia virus early/late promoter PmH5 in the MVA vector plasmid pL-W73_SARS-2-N and integrated by homologous recombination into the intergenomic regions of the open reading frames 069R and 070L of the MVA genome ([36]; Figure 1A). For the generation of the single MVA-SARS-2-N virus (MVA-N), we used the non-recombinant MVA virus (isolate F6 [9,48]) as the backbone virus. For the generation of the multivalent MVA-SARS-2-ST/N (MVA-ST/N) expressing the SARS-2-N and the SARS-2-ST in the same virus, we used the MVA-SARS-2-ST virus (MVA-ST) [10] as the backbone virus and integrated the gene encoding sequence for the N-protein as described above (Figure 1A). 

Genetic integrity of the recombinant MVAs was confirmed by the PCR analysis of viral DNA, demonstrating the site-specific insertion of the gene sequences from N-protein and S-protein of SARS-CoV-2 as well as the absence of non-recombinant MVA or MVA-ST and the proper removal of the GFP-marker gene (Appendix A). Furthermore, genetic stability of the recombinant MVA viruses was confirmed by PCR targeting the six major deletion sites and the C7L gene locus of MVA (Appendix A). The recombinant viruses replicated efficiently in chicken embryo fibroblasts (CEFs) or DF-1 cells, but not in the human HaCat cells (Figure 1B). To further characterize the expression pattern of the recombinant SARS-2-N and SARS-2-ST, total cell lysates from CEF cells infected with MVA-N or the multivalent MVA-ST/N were analyzed by a Western blot. The mouse monoclonal antibody directed against the N-protein revealed one prominent protein band that migrated with molecular masses of ~49.5 kDa (Figure 1C). In general, for both recombinant MVA viruses, N-protein synthesis was first detected at 8 h post infection (hpi) and synthesis further increased over 24 to 48 hpi (Figure 1C). For lysates from cells infected with the multivalent MVA-ST/N virus, we detected an additional band at ~190 kDa when using a monoclonal antibody directed against the S2-subunit of the S-protein (Figure 1C). The Western blot analysis directly comparing the expression pattern of MVA-N and MVA-ST/N at 24 hpi confirmed these results (Figure 1D). Next, we used immunofluorescent staining with SARS-2-N-specific primary antibodies and SARS-2-S2-specific primary antibodies against the S2-domain of the S-protein to analyze the expression of the SARS-CoV-2 proteins on the cell surface as well as the trafficking upon MVA-N or MVA-ST/N infection in Vero cells (Figure 1E). In control cultures infected with non-recombinant MVA or cells that remained un-infected, we did not observe any background staining. Upon infection with the MVA-N or MVA-ST/N, we observed a reticular pattern with a juxtanuclear accumulation of N-protein. To further investigate the distribution patterns of SARS-2-ST, we also performed immunostaining of infected cells using the S2-specfic antibody. Here, we specifically detected the S-protein only in cells infected with MVA-ST/N, indicating abundant S-protein synthesis (Figure 1E).

### 3.2. Single Vaccination with MVA-ST/N Induces Robust Protection against a Lethal SARS-CoV-2 Challenge Infection in K18-hACE2 Mice

To assess the immunogenicity and efficacy of the MVA-ST, MVA-N, and MVA-ST/N candidate vaccines after a single shot vaccination, we used the transgenic K18-hACE2 mice, a lethal mouse model for SARS-CoV-2 infection. We vaccinated the mice once with a standard dosage of 1 × 10^8^ PFU of MVA-ST (*n* = 7), MVA-N (*n* = 8), MVA-ST/N (*n* = 8), or MVA (*n* = 4) as a vector control by the intramuscular route. Additionally, one group of mice received only PBS as mock vaccination (*n* = 4). A total of 4 weeks after prime vaccination, all mice were intranasally (i.n.) infected with a lethal dose of a 3.6 × 10^4^ TCID_50_ of SARS-CoV-2 (BavPat1), followed by daily body weight measurements and clinical scoring. A total of 8 days post infection, all mice were euthanized, and organ samples were taken to analyze viral load and immune responses. Mice belonging to the control groups (PBS, MVA) started to lose weight around 3 days post infection (3 dpi) with a mean of 4% weight loss for PBS and 2.3% weight loss for MVA of initial body weight. Additionally, MVA-N-vaccinated mice also lost weight starting at 3 dpi with a mean of 3.1% weight loss from initial body weight (Figure 2A). They suffered from disease by showing SARS-CoV-2-specific clinical symptoms and showed further weight loss with a mean of 15.3% weight loss (PBS), 23.1% weight loss (MVA), and 18.4% weight loss (MVA-N) of initial body weight and all succumbed to infection or had to be euthanized by day 6. 

MVA-N-vaccinated mice, very comparable to the control-vaccinated mice, suffered from substantial weight loss, comprising a mean of 18.4% weight loss from the initial body weight, and SARS-CoV-2-specific symptoms on day 6 post infection (Figure 2A–C). For the control-vaccinated animals, we detected high levels of SARS-CoV-2 RNA in the upper respiratory tract as analyzed by RT-qPCR of oropharyngeal swabs on day 4 and the day of death (final) with a mean titer of 5.9 × 10^3^ RNA copy numbers/μL (PBS) and 4.8 × 10^3^ RNA copy numbers/μL (MVA) on day 6 post challenge (Figure 2F). Control mice also showed high titers of the infectious virus in the lungs with a mean titer of a 7.2 × 10^3^ TCID_50_/mL (PBS) and a 3.5 × 10^4^ TCID_50/_mL (MVA) and even higher titers in the brain with a mean titer of a 9.9 × 10^6^ TCID_50_/mL (PBS) and a 1.9 × 10^7^ TCID_50_/mL (MVA) as analyzed by TCID_50_ (Figure 2D). These data were further confirmed by the RT-qPCR analysis targeting the gene of the RNA-dependent RNA polymerase (RDRP) of SARS-CoV-2 with mean titers of 3.3 × 10^6^ RNA copy numbers/μL (lung) and 4.4 × 10^7^ RNA copy numbers/μL (brain) for the PBS group as well as 3.2 × 10^6^ RNA copy numbers/μL (lung) and 8.4 × 10^7^ RNA copy numbers/μL (brain) for the MVA group (Figure 2E). In MVA-N-vaccinated mice, we detected high levels of SARS-CoV-2 RNA in the upper respiratory tract shown by a mean of 3.4 × 10^3^ RNA copy numbers/μL on day 4 post infection and a mean of 7 × 10^3^ RNA copy numbers/μL on the day of death (final) (Figure 2F). MVA-N-vaccinated mice showed substantial infectious SARS-CoV-2 titers in the lungs (mean of 1.6 × 10^4^ TCID_50/_mL) and in the brain (mean of 4.7 × 10^7^ TCID_50/_mL) (Figure 2D). These results were further confirmed by the RT-qPCR analysis with a mean of 1.1 × 10^6^ RNA copy numbers/μL for the lung and 1.8 × 10^8^ RNA copy numbers/μL for the brain (Figure 2E). All MVA-ST- and MVA-ST/N-vaccinated mice seemed to be fully protected against the challenge infection without any body weight loss or any obvious clinical signs (Figure 2A–C). We did not detect any titers of the infectious virus in the lungs and brains of MVA-ST- and MVA-ST/N-vaccinated mice (Figure 2D). These data were further confirmed by data from viral RNA in the lungs (MVA-ST with a mean of 5.8 × 10^3^ RNA copy numbers/μL and MVA-ST/N with a mean of 8.3 × 10^3^ RNA copy numbers/μL), in the brains (MVA-ST with a mean of 3 × 10^4^ RNA copy numbers/μL and MVA-ST/N with a mean of 1 × 10^3^ RNA copy numbers/μL), and in oropharyngeal swabs (MVA-ST with a mean of 3.6 × 10^1^ RNA copy numbers/μL and MVA-ST/N with a mean of 1 × 10^1^ RNA copy numbers/μL for final swabs) (Figure 2E,F). We did not detect obvious titers of neutralizing antibodies (nABs) in control- and MVA-N-vaccinated animals before the challenge infection. After SARS-CoV-2 infection, control-vaccinated mice and MVA-N-vaccinated mice mounted detectable levels of SARS-CoV-2 nAB (mean titer of 1:1860 PRNT_50_ for PBS, mean titer of 1:187.5 PRNT_50_ for MVA, and mean titer of 1:296.25 PRNT_50_ for MVA-N-vaccinated animals) (Figure 3A). 

Single vaccination with MVA-ST and MVA-ST/N induced the production of nAB against SARS-CoV-2 before the challenge and 18 days post vaccination with a mean titer of 1:788 PRNT_50_ for MVA-ST and 1:2640 PRNT_50_ for MVA-ST/N. These titers were further boosted after challenge infection on day 28 (Figure 3A). Here, mice mounted a mean titer of 1:5074 PRNT_50_ for MVA-ST and 1:5880 PRNT_50_ for MVA-ST/N-vaccinated animals. Before challenge infection, we also did not detect substantial titers of N-specific antibodies in control- or MVA-ST-vaccinated mice. In contrast, MVA-ST/N-vaccinated mice showed significant titers after vaccination with a mean titer of 1:300, which further increased after challenge infection (mean titer of 1:762) (Figure 3B). For the characterization of T cells after challenge infection at the end of the experiment, the IFN-γ-based ELISpot assay from the lung and spleen and flow cytometry using SARS-CoV-2-specific tetramer staining were performed from blood, lungs, spleens, and brains (Figure 3C–H, Appendix A). The MVA-ST and MVA-ST/N vaccination induced substantial levels of S_V8L_-specific T cells (CD8^+^ CD44^+^ S_V8L_^+^) in the blood (mean of 9 × 10^2^ counts/mL for MVA-ST, mean of 2.2 × 10^3^ counts/mL for MVA-ST/N) analyzed by flow cytometry (Figure 3F). For the characterization of S_V8L_-specific T cells (CD8^+^ CD44^+^ S_V8L_^+^) in the spleen, we detected a mean of 3.9 × 10^2^ counts/10^6^ splenocytes for MVA-ST and a mean of 5.5 × 10^2^ counts/10^6^ splenocytes for MVA-ST/N by flow cytometry (Figure 3C). We observed significant S_V8L_-specific T cell levels for MVA-ST (mean of 8.7 × 10^2^ of IFN-γ SFCs/10^6^ splenocytes)- and MVA-ST/N (mean of 2.5 × 10^3^ of IFN-γ SFCs/10^6^ splenocytes)-vaccinated animals shown by ELISpot (Figure 3G). These data were further confirmed in the lung with a mean of 2.9 × 10^3^ counts/10^6^ lung cells (flow cytometry) and 7.2 × 10^3^ of IFN-γ SFCs/10^6^ lung cells (ELISpot) for MVA-ST as well as 2.6 × 10^3^ counts/10^6^ lung cells (flow cytometry) and 9.6 × 10^3^ of IFN-γ SFCs/10^6^ lung cells (ELISpot) for MVA-ST/N (Figure 3D,H). Increased titers of N_219_-specific T cells (CD8^+^ CD44^+^ N_219_^+^) were found in brains of MVA-N-vaccinated animals with mean titers of 1.3 × 10^1^ counts/10^6^ of brain cell suspension (Figure 3E). N_219_-specific T cells (CD8^+^ CD44^+^ N_219_^+^) were rarely found in the lungs of control animals with a mean of 7 × 10^1^ counts/10^6^ lung cells for PBS and a mean of 8.8 × 10^1^ counts/10^6^ lung cells for MVA-vaccinated mice. Vaccination with MVA-ST, MVA-N, and MVA-ST/N revealed higher titers of N_219_-specific T cells (CD8^+^ CD44^+^ N_219_^+^) in the lungs with a mean of 8.5 × 10^1^ counts/10^6^ lung cells for MVA-ST, a mean of 1.5 × 10^2^ counts/10^6^ lung cells for MVA-N, and a mean of 1 × 10^2^ counts/10^6^ lung cells for MVA-ST/N-vaccinated mice (Figure 3D). MVA_B8R_-specific T cells (CD8^+^ CD44^+^ B8R_20–27_^+^) could not be found in unvaccinated animals (PBS), whereas marginal amounts were found in MVA- and MVA-N-vaccinated mice. More abundant amounts of MVA_B8R_-specific T cells were found in MVA-ST- and MVA-ST/N-vaccinated mice with significant titers in the spleen, lung, and blood of MVA-ST/N-vaccinated animals, shown by the flow cytometry (Appendix A–D) and ELISpot analysis (Appendix A). These results were further confirmed by pathohistological results from the lungs and the brains (Figure 4).

Here, control-vaccinated mice (PBS and MVA) and MVA-N-vaccinated mice revealed significant amounts of a SARS-2-S antigen in the lungs (mean of 15.7% SARS-2-S-positive (+) cells for PBS, mean of 8.6% SARS-2-S+ cells for MVA, and mean of 5.1% SARS-2-S+ cells for MVA-N). For MVA-ST- and MVA-ST/N-vaccinated mice, the amount of an S-antigen in the lungs was significantly reduced with a mean of 2.8% SARS-2-S+ cells for MVA-ST and a mean of 2.1% SARS-2-S+ cells for MVA-ST/N (Figure 4A). The quantitative evaluation of the viral antigen staining in the brains revealed a large area, coalescing to diffuse areas with numerous N-positive neurons, located in the cerebrum and brain stem. The control-vaccinated mice reached a mean of 1.4% SARS-2-N+ area for PBS and 2.0% SARS-2-N+ area for MVA-vaccinated animals. MVA-N-vaccinated animals also mounted substantial N-positive areas (mean of 1.3% SARS-2-N+ area). For MVA-ST- and MVA-ST/N-vaccinated animals, these areas were significantly reduced with a mean of 0.004% SARS-2-N+ area for MVA-ST and 0.009% SARS-2-N+ area for MVA-ST/N vaccination (Figure 4B). 

### 3.3. MVA-ST/N Emergency Vaccination 2 Days before a Lethal SARS-CoV-2 Challenge Infection Resulted in Delayed Outcome of Disease and Death

Based on the results from single vaccination, we further aimed to test the MVA-ST/N candidate vaccine for its capacity to rapidly protect after single vaccination. Mice were vaccinated (i.m.) with MVA-ST/N (*n* = 7), MVA as a vector control (*n* = 7), or PBS as a mock control (*n* = 7). Two days after the emergency vaccination, all mice were challenged (i.n.) with a lethal dose of a 3.6 × 10^4^ TCID_50_ of SARS-CoV-2 (BavPat1) as established before. Body weight changes as well as the development of clinical symptoms were measured daily. Based on the cumulative clinical score, the mice were euthanized, and organ samples were prepared for the characterization of viral load and immune responses. All animals (PBS, MVA, MVA-ST/N) started to lose body weight at 3 days post infection (3 dpi) with a mean of 1.1% weight loss for PBS, a mean of 0.3% weight loss for MVA, and a mean of 3.9% weight loss for MVA-ST/N-vaccinated mice, followed by further significant body weight loss until 5 dpi (Figure 5A). 

Control-vaccinated mice showed a mean weight loss of 20.3% (PBS) and 19.9% (MVA) from the initial body weight while the MVA-ST/N-vaccinated animals had a mean weight loss of 16.1% of initial body weight on day 6, which remained stable until 8 days post infection (Figure 5A). Additionally, all mice started to develop clinical signs, like ruffled fur, enhanced breathing, and diminished activity at 4 dpi with a cumulative score ranging from 1 to 3. Mice vaccinated with MVA-ST/N showed a milder clinical disease outcome as measured by the clinical score sheet, as established before, five to six days post infection compared to the control-vaccinated mice (Figure 5C). All control-vaccinated mice died or had to be euthanized by day 6 due to progressed body weight loss and clinical signs culminating in a total clinical score of 7; 8.71% (5 out of 7) of the MVA-ST/N-vaccinated animals showed a longer survival rate than the control groups and were euthanized at 7 dpi (*n* = 2) or at the end of the experiment at 8 dpi (*n* = 3) with a cumulative score ranging from 6 to 7 (Figure 5A–C). Oropharyngeal swabs were taken four days post infection (4 dpi) and on the day of death and viral RNA titers were analyzed by RT-qPCR. Substantial viral shedding was measured in the oropharyngeal swabs at day 4 for all animals (mean of 5.5 × 10^3^ RNA copy numbers/μL for PBS, mean of 1.9 × 10^3^ RNA copy numbers/μL for MVA, and mean of 2.2 × 10^3^ RNA copy numbers/μL for MVA-ST/N). Significantly reduced viral shedding was measured in oropharyngeal swabs of MVA-ST/N-vaccinated animals on the day of death (final) with a mean of 6.7 × 10^2^ RNA copy numbers/μL compared to 3.9 × 10^3^ RNA copy numbers/μL for PBS and 1.4 × 10^3^ RNA copy numbers/μL for MVA (Figure 5F). Lungs and brains were analyzed for viral load by TCID_50_ and the RT-PCR analysis. All mice from the control groups showed high titers of infectious SARS-CoV-2 in lungs and brains (median of 5 × 10^4^ TCID_50/_mL (lung) and 5 × 10^6^ TCID_50/_mL (brain) for PBS, 5 × 10^4^ TCID_50/_mL (lung) and 5 × 10^6^ TCID_50/_mL (brain) for MVA) analyzed by the TCID_50_ assay (Figure 5D). These data were further confirmed by RT-qPCR for viral RNA load with a mean of 1.2 × 10^7^ RNA copy numbers/μL (lung) and a median of 1.3 × 10^8^ RNA copy numbers/μL (brain) for PBS as well as a mean of 2.2 × 10^7^ RNA copy numbers/μL (lung) and a median of 1.2 × 10^8^ RNA copy numbers/μL (brain) for MVA (Figure 5E). MVA-ST/N vaccination in the lung resulted in the absence of infectious SARS-CoV-2, except for one mouse (Figure 5D). This was further confirmed for the RT-qPCR analysis where significantly lower titers of viral RNA in the lung were observed with a mean titer of 3.7 × 10^6^ RNA copy numbers/μL (Figure 5E). In the brains of the MVA-ST/N-vaccinated mice, we detected substantially reduced titers of the infectious virus (median of 8.9 × 10^5^ TCID_50/_mL) as well as viral RNA of SARS-CoV-2 (median of 7.6 × 10^7^ copy numbers/μL) (Figure 5D,E). For the control-vaccinated mice, we did not detect any obvious neutralizing antibody (nAB) titers or N-specific antibodies at the end of the experiment. In contrast, MVA-ST/N vaccination 2 days before SARS-CoV-2 challenge infection induced increased titers of nAB with a mean titer of 1:3600 PRNT_50_ (Figure 6A) and N-specific antibodies with a mean of 1:2475 (Figure 6B).

Splenocytes and lung cells were prepared and blood was taken after challenge infection at the end of the experiment to analyze the presence of T cells. The characterization of specific T cells in the spleen and lung was performed as established before using ELISpot and a flow cytometry analysis (Appendix A). In addition, we also used pools of overlapping peptides comprising the whole N-protein (N Pool) to further characterize the N-specific T cells by the ELISpot assay. The phenotypic characterizations of T cells in the blood, spleen, and lung were additionally performed using the flow cytometry analysis. In the control-vaccinated mice, we detected levels of S_V8L_-specific T cells in the spleen (mean of 4.3 × 10^1^ of IFN-γ SFCs/10^6^ splenocytes for PBS, mean of 6 × 10^1^ of IFN-γ SFCs/10^6^ splenocytes for MVA) and in the lung (mean of 4.4 × 10^2^ of IFN-γ SFCs/10^6^ lung cells for PBS, mean of 4.8 × 10^2^ of IFN-γ SFCs/10^6^ lung cells for MVA) by ELISpot (Figure 6C,D). MVA-ST/N-vaccinated mice mounted substantial levels of S_V8L_-specific T cells in the spleen with a mean of 2.5 × 10^2^ of IFN-γ SFCs/10^6^ splenocytes (Figure 6C) and even more elevated titers in the lung with a mean of 8 × 10^3^ of IFN-γ SFCs/10^6^ lung cells (Figure 6D). Of note, levels of N_219_-specific T cells were neglectable for all mice. For a more detailed characterization of the specific T cell subsets, blood, splenocytes, and lung cells were stained with CD3-, CD4-, CD8-, and CD44-specific antibodies followed by staining with a Live/Dead Kit, and analyzed by flow cytometry (Figure 6E–G). In control mice, we detected activated CD4 T cells (CD4^+^ CD44^+^) in the spleen with a mean of 1 × 10^5^ counts/10^6^ splenocytes for PBS and 8.7 × 10^4^ counts/10^6^ splenocytes for MVA-vaccinated animals. For activated CD8 (CD8^+^ CD44^+^) T cells, we detected a mean of 3.1 × 10^4^ counts/10^6^ splenocytes in PBS and 2 × 10^4^ counts/10^6^ splenocytes in MVA (Figure 6E). MVA-ST/N-vaccinated mice mounted significantly higher levels of activated CD4 (CD4^+^ CD44^+^) and activated CD8 (CD8^+^ CD44^+^) T cells in the spleen with a mean titer of 1.5 × 10^5^ counts/10^6^ splenocytes for CD4 and 1 × 10^5^ counts/10^6^ splenocytes for CD8 T cells. Lungs of control animals contained activated CD4 (CD4^+^ CD44^+^) and activated CD8 (CD8^+^ CD44^+^) T cells with mean titers of 5.6 × 10^3^ counts/10^6^ lung cells for CD4 and 8.3 × 10^3^ counts/10^6^ lung cells for CD8 T cells in PBS mice as well as mean titers of 1.3 × 10^4^ counts/10^6^ lung cells for CD4 and 2 × 10^4^ counts/10^6^ lung cells for CD8 T cells in MVA-vaccinated animals (Figure 6F). MVA-ST/N-vaccinated animals showed significantly increased numbers of activated CD4 (CD4^+^ CD44^+^) T cells, with a mean of 3 × 10^4^ counts/10^6^ lung cells, and increased levels of activated CD8 (CD8^+^ CD44^+^) T cells, with a mean of 7.6 × 10^4^ counts/10^6^ lung cells, in the lung (Figure 6F). One MVA-ST/N-vaccinated animal showed a high activated T cell count in the blood after challenge infection, mounting 2.4 × 10^4^ counts/mL for CD4 T cells and 4.3 × 10^4^ counts/mL for CD8 T cells (Figure 6G). Unvaccinated mice (PBS) did not show any MVA_B8R_-specific T cells, neither in the spleen nor in the lung nor in blood, analyzed by flow cytometry. We could confirm specific T cells targeted against the vaccine vector MVA (CD8^+^ CD44^+^ B8R_20–27_^+^) in control-vaccinated mice (MVA) with significant amounts in the spleen (median of 1.2 × 10^3^ counts/10^6^ splenocytes), in the lung (mean titers of 2.3 × 10^3^ counts/10^6^ lung cells), and in the blood (median of 3.6 × 10^2^ counts/mL) six days after challenge infection (Appendix A–C), analyzed by flow cytometry. Similarly, significant MVA_B8R_-specific T cell titers (CD8^+^ CD44^+^ B8R_20–27_^+^) were observed in the lung (mean of 1.6 × 10^3^ counts/10^6^ lung cells) as well as substantial titers in the spleen (median of 2.5 × 10^2^ counts/10^6^ splenocytes) of MVA-ST/N-vaccinated mice (Appendix A). We also confirmed substantial amounts of MVA_B8R_-specific T cells in spleens and lungs of MVA- and MVA-ST/N-vaccinated mice by ELISpot analysis with a median of 2.9 × 10^2^ of IFN-γ SFCs/10^6^ splenocytes and 2.9 × 10^3^ of IFN-γ SFCs/10^6^ lung cells for MVA as well as 2.5 × 10^1^ of IFN-γ SFCs/10^6^ splenocytes and 1.7 × 10^3^ of IFN-γ SFCs/10^6^ lung cells for MVA-ST/N-vaccinated mice (Appendix A). 

Pathohistological results of lungs and brains further confirmed these results (Figure 7). 

Sections of lungs and brains were analyzed using immunohistochemistry (IHC) targeting the SARS-2-S protein (lung) or the SARS-2-N protein (brain). Here, unvaccinated mice (PBS) and control-vaccinated mice (MVA) showed significant immunolabeling for SARS-2-S in the lungs shown by a mean of 21.3% SARS-2-S+ cells for PBS and 20.8% SARS-2-S+ cells for MVA. Lungs of MVA-ST/N-vaccinated animals mounted significantly less of a signal for the S-antigen as shown by a mean of 8.9% SARS-2-S+ cells (Figure 7A). An immunohistological analysis of the brain sections revealed a large area, coalescing to diffuse areas with numerous SARS-2-N+ neurons in brains of control-vaccinated mice with a mean of 4.4% SARS-2-N+ area for PBS and a mean of 3.5% SARS-2-N+ area for MVA (Figure 7B). MVA-ST/N-vaccinated mice showed significantly reduced immunolabeling for the N-antigen in the brain with a mean of 0.7% SARS-2-N+ area (Figure 7B).

## 4. Discussion

This study demonstrates that single vaccination with a multiantigen MVA vaccine co-expressing the SARS-CoV-2 S- and N-protein (MVA-ST/N) 28 days before a lethal SARS-CoV-2 infection induced robust respiratory and neurological protection in K18-hACE2 mice. Moreover, MVA-SARS-2-ST/N (MVA-ST/N) vaccination also had a beneficial effect on the outcome of disease in these mice when applied just 2 days before the lethal SARS-CoV-2 challenge infection. In addition to previous data demonstrating a strong cross-protective capacity [24,26,28], our data also indicate a rapid protective potential of such a multiantigen MVA-COVID-19 vaccine candidate. These data will contribute to the development of improved vaccination strategies against emerging pathogens exemplified by SARS-CoV-2. The experience with the COVID-19 pandemic and the still ongoing spread of SARS-CoV-2 indicate the need for vaccines that rapidly protect against an emerging pathogen. To further improve vaccine efficacy with regard to broader-protective and rapidly protective immunizations, several approaches have been undertaken. For mRNA vaccines, formulation and dosages have been optimized to result in a stronger activation of immune responses to improve the protective efficacy [49,50,51,52,53,54]. Several other approaches evaluated the effect of heterologous prime-boost regimens for the activation of broader-reactive and more robust immune responses [55,56,57,58]. For the activation of a more cross-reactive immune response also mediating protection against multiple variants of SARS-CoV-2, the combination of the S-protein with the more conserved N-protein has been successfully evaluated in several different vaccine approaches [25,26,27,28]. Very recent studies confirmed that MVA-based vaccine candidates co-expressing SARS-CoV-2-S and -N induced a strong activation of a humoral immune response that neutralized several different SARS-CoV-2 variants also including viral variants of concern (VOCs). These multivalent MVA vectors induced robust protection against different VOCs when tested in the Syrian hamster and non-human primates (NHPs) [10,23,24,28]. As a mechanism of the improved cross-reactive efficacy, the higher titers of neutralizing antibodies (nABs) mediated by the combination of S- and N-protein have been hypothesized [28]. Based on these data, we hypothesized that a combination of S- and N-protein expressed by MVA could also rapidly protect against lethal SARS-CoV-2 challenge infection. Robust protection after a single vaccination might also be a promising strategy since repeated vaccination with first-generation S-based COVID-19 vaccines has been confirmed to result in waning immunity [13,59]. In previous studies, we already confirmed the rapid protective capacity of a single MVA vaccination against lethal orthopoxvirus challenge infection [60,61]. Here, a single MVA vaccination robustly protected mice against lethal VACV and ECTV challenge infection [61]. A more detailed analysis of the immune responses indicated a superior role of the cellular immune response for rapid protection 2 days after initial vaccination [60,61]. Confirming previous studies [23], we showed that a single MVA-ST/N vaccination resulted in robust protection against the lethal respiratory SARS-CoV-2 challenge 28 days later. An interesting aspect of this single vaccination study was that the MVA-based vaccines expressing only one SARS-CoV-2 protein, either the S-protein or the N-protein, resulted in reduced protective capacity. For the monovalent MVA-SARS-2-ST candidate vaccine (MVA-ST), mice were in principle protected before death and disease. Even more obvious was the altered outcome of MVA-SARS-2-N (MVA-N) vaccination as we did not detect any protective effect. MVA-N-vaccinated animals suffered from morbidity and mortality, very comparable to the disease outcome experienced in the control-vaccinated animals (mock and empty MVA). This altered outcome of protection correlated with the viral load in the upper and lower respiratory tract and in the highly susceptible brain. Already on day 4, the viral shedding, as measured by oropharyngeal swabs, was more obviously reduced in the MVA-ST/N-vaccinated animals followed by the MVA-ST-vaccinated animals and no reduction in viral shedding as measured in the MVA-N- and control-vaccinated animals. On the day of death (day 8 for MVA-ST and MVA-ST/N, day 6 for MVA-N and control animals), the viral shedding was significantly reduced in the MVA-ST/N- and MVA-ST-vaccinated animals while still no effect was visible in the other animals. This pattern for SARS-CoV-2 load was further confirmed for the lower respiratory tract. Of note, we did not detect any infectious virus in the brain of one MVA-N-vaccinated animal. However, no other correlation between the absence of the virus and the clinical score could be seen. In line with the results from viral loads, the quality and quantity of the immune responses were also different between the different groups. Only MVA-ST- or MVA-ST/N-vaccinated mice mounted virus-neutralizing antibodies before and after SARS-CoV-2 challenge infection. Titers of neutralizing antibodies induced by the MVA-SARS-2-ST candidate vaccine after single-shot vaccination are comparable to the titers of neutralizing antibodies measured after prime-boost vaccination in the K18-hACE2 mouse model [10]. Here, an advantage of the multivalent MVA-ST/N vaccine over the MVA-ST vaccine to stimulate higher titers of nAb was shown. The MVA-ST/N vaccine also induced N-binding antibodies to substantially higher titers compared to the MVA-N vaccine. Importantly, antibodies to both the S- and N-antigens detected in SARS-CoV-2-infected humans have been associated with a mild-asymptomatic disease outcome after SARS-CoV-2 infection [62]. Here, it will be of interest for future studies to also characterize the titers of S-binding antibodies induced by the MVA-SARS-2-ST/N vaccine. Moreover, in these humans, clearance of SARS-CoV-2 has been confirmed to be more rapid and long-lasting [63]. These data indicated a beneficial impact of the combination of S- and N-protein. This combinatory effect might also explain the failed protection induced by the MVA-N vaccine. This might be further explained with the absence of neutralizing antibodies mainly induced by the SARS-2-S antigen [7,8]. This is further underlined by the data from the T cell characterization. Here, MVA-ST/N vaccination induced significantly improved levels of SARS-2-S-specific T cells compared to the MVA-ST-, MVA-N-, and control-vaccinated animals. Of note, increased levels of activated CD8+ T cells for MVA-ST/N vaccination have been confirmed in the blood, which is indicative of peripheral CD8+ T cells, as well as in the lung as the main target organ for SARS-CoV-2 infection. Moreover, the spleen, as an important immunological organ for the maturation of T cells, also mounted improved levels of S-specific T cells in MVA-ST/N-vaccinated animals. The very balanced activation of S- and N-specific humoral immune responses paired with robust levels of S-specific T cells might be an explanation for the robust and improved protection after a single immunization with MVA-ST/N 4 weeks earlier. A previous study confirmed the advantage of a balanced activation of cellular immune responses for the outcome of the protection of mice that had been vaccinated with an MVA-based candidate vaccine against the Ebola virus [64]. While we did not detect N-specific T cells in the blood, the spleen, and the lung, we detected substantial levels of N-specific CD8+ T cells in the brain of MVA-N-vaccinated mice. No N-specific CD8+ T cells were detected in control-vaccinated and MVA-ST- and MVA-ST/N-vaccinated mice. In previous studies, N-specific T cells have been hypothesized to induce a broader protective potential than S-specific T cells due to the higher conservation of the N-protein [34]. However, the impact of N-specific T cells for a more rapid protective capacity has not been evaluated in more detail. In that context, the role of T cells in the brain also needs to be evaluated in more detail, since the brain has been confirmed to be an immunoprivileged organ and several studies confirmed immunopathology induced by specific immune responses [65,66,67,68]. In our study, we did not detect indications of vaccine-induced immunopathology. However, these results need to be characterized in more detail in future studies to evaluate the impact of the N-specific T cells for the outcome of protection, and also disease manifestation. In previous studies, the activation of lung-specific T cells has been considered to be essentially required to robustly protect against lethal SARS-CoV-2 challenge infection [69,70]. Of note, to evaluate the role of T cells induced by single vaccination with the different MVA-SARS-2 candidate vaccines in more detail and define possible immunological correlates of protection, future studies characterizing the cellular immune responses without challenge infection to differentiate between the vaccination and other immune response reactivations are warranted. The robust protection of the MVA-ST/N vaccine was confirmed since we did not detect SARS-CoV-2 immunohistologically nor with the highly sensitive RT-qPCR analysis. In previous studies using the highly susceptible K18-hACE2 mouse model, the robust activation of the humoral immune responses alone was not sufficient to prevent initial SARS-CoV-2 infection, which induced minimal lung lesions and detectable virus load by qRT-PCR. In contrast, the activation of antibodies and T-cell-mediated robust protection against ancestral SARS-CoV-2 as well as variants of concern was noted [70,71,72]. Thus, that no initial SARS-CoV-2 infection has been measured could be associated with the advanced activation of the cellular immune responses. Of note, the strong activation of nAb can also be associated with reduced viral shedding. This is promising already after a single vaccination. In summary, the activation of improved S-specific antibodies and T cells might be a possible explanation for the improved outcome of protection after MVA-ST/N vaccination. While no immunological mechanism was identified, we hypothesize that the S-specific immune responses are essentially required to mediate protection since we did not detect protective efficacy after MVA-N vaccination. Based on the slightly improved effects for immunogenicity and efficacy seen in the MVA-SARS-2-ST/N vaccine in the single vaccination approach, we aimed to evaluate the MVA-SARS-2-ST/N vaccination in a shortened interval. Using an emergency vaccination regimen, the outcome of morbidity and mortality in MVA-ST/N was delayed compared to the control-vaccinated mice with a mean time of death of 7 compared to 6. This was most obvious in the significant reduced SARS-CoV-2 shedding from the upper respiratory tract and reduced levels of SARS-CoV-2 in the lung of MVA-ST/N-vaccinated mice. Again, we confirmed that reduced viral shedding in the emergency vaccination regimen is associated with robust titers of neutralizing and N-specific antibodies. This is in line with previous data from studies in humans confirming that a strong activation of SARS-CoV-2-specific antibodies correlated with reduced viral shedding. Moreover, a very rapid activation of antibodies has been associated with more rapid clearance and reduced clinical symptoms [73,74]. However, despite the effect on the respiratory tract, there was no obvious effect on neuroprotection in the emergency vaccination regimen since we detected substantial levels of SARS-CoV-2 in the brain. Of note, immunohistological results indicate a beneficial effect since there was a significant reduction in a viral antigen in the brain of MVA-ST/N-vaccinated mice. To correlate the delayed outcome of morbidity and mortality with the activation of specific immune responses, we detected an increased activation of virus-neutralizing antibodies in several of these mice as well as N-specific antibodies in one mouse. In addition, we also confirmed levels of S-specific T cells in the lung and in the spleen. Of note, in the blood, levels of activated T cells after challenge infections were not significantly different compared to the control-vaccinated animals. Since it has been demonstrated that the systemic SARS-CoV-2 spread can also be mediated via the blood [75,76,77], we hypothesize a role of the blood T cells for the prevention of SARS-CoV-2 infection within the brain. Since the brain manifestation is a non-COVID-19-disease-specific outcome of the K18-hACE2 mouse model and associated to the artificial expression of the human ACE2 receptor molecule in the brain [78], it might be interesting to test the emergency vaccination regimen in another animal model without the dominant brain manifestation and neurological outcome. Whether the efficacy of emergency MVA-ST/N vaccination can be further enhanced by using increased dosages or other immunization routes remains to be addressed in future studies. Moreover, future studies evaluating the early activation and kinetics of T cell responses without the SARS-CoV-2 challenge infection, which might serve as a boost effect, should be characterized. While we did not assess the effect of emergency vaccination using the MVA-ST and MVA-N vaccine candidates, we demonstrated that there is no or reduced protective efficacy compared to the multiantigen MVA-ST/N candidate vaccine using a single 28-day vaccination regimen. For this, we hypothesize that this would be similar in the emergency vaccination approach, and we only tested the emergency vaccination approach for the MVA-ST/N vaccine. So far, no detailed studies have been published that confirmed the protective efficacy of a single MVA vaccine expressing the N-protein in the context of coronaviruses. However, this has been confirmed for innovative vaccination strategies against the influenza virus. Here, several studies confirmed a robust protection of an MVA vaccine expressing influenza virus NP proteins also against different strains of the influenza virus [17,19]. These studies included vector vaccine approaches as well as protein-based vaccines and mRNA-based vaccines [17,19,79]. Here, the broadly reactive protective outcome was associated with a robust activation of T cells in the lung and in the blood. Future studies will be required to analyze the role of the blood T cells in vaccine-induced protection for the COVID-19 approach in more detail. In that context, it will be of significant importance to also evaluate the rapid protective capacity against different variants of SARS-CoV-2, also including VOCs. With regard to a long-term protective efficacy, which will also contribute to minimize the risk for the emergence of new VOCs, future studies also evaluating the immunogenicity and efficacy at late immunization schedules are important and interesting. In summary, these data further highlight the promising potential of a multivalent vaccine co-expressing S- and N-protein as confirmed in a lethal mouse model for COVID-19. Using a multivalent MVA vaccine co-expressing the S- and N-protein for efficacy testing in mice, we also overcome the difficulties of simply mixing both vaccine candidates MVA-N and MVA-ST, which is not possible due to the limiting volumes for vaccination in mice.

## 5. Conclusions

In conclusion, our results provided preliminary data on more rapidly protective vaccination approaches for COVID-19. Here, we confirmed that a single MVA-ST/N vaccination robustly protects highly susceptible K18-hACE2 mice against a lethal SARS-CoV-2 challenge infection 28 days later while the emergency vaccination after 2 days was only partially protective and did not prevent the outcome of disease and death. Of note, this morbidity and mortality could be associated with an effect on SARS-CoV-2 infection in the upper and lower respiratory tract and no effect on the neuroprotection. Based on these preliminary data, more detailed studies evaluating the immune responses associated with more rapid protective vaccination should be initiated. In addition, future studies evaluating the rapid protective efficacy of the MVA-ST/N vaccine against different SARS-CoV-2 variants need to be tested. 

## Figures and Tables

**Figure 1 viruses-16-00417-f001:**
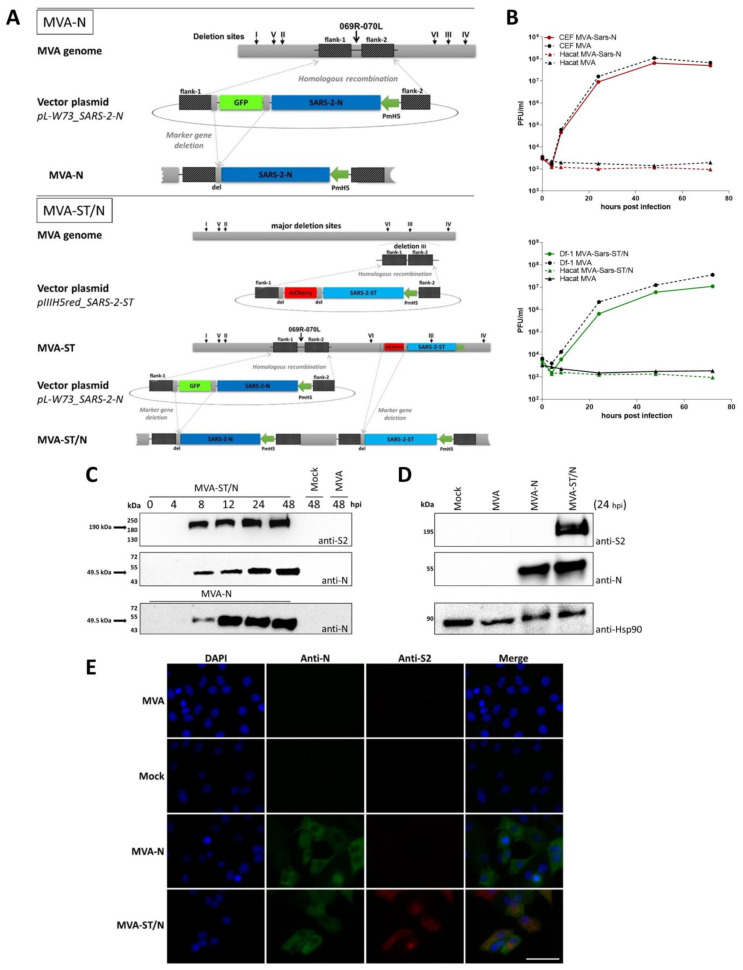
Construction and in vitro characterization of MVA-N and MVA-ST/N. (**A**) Schematic diagram of the MVA genome with the major deletion sites I to VI. MVA-SARS-2-N (MVA-N): The intergenomic region 069R and 070L of the MVA genome was targeted by inserting the gene sequence encoding the N-protein of SARS-CoV-2 from the virus isolate Wuhan HU-1 under the control of the vaccina virus promoter PmH5. Repetitive sequences served to remove the GFP-marker gene by intragenomic homologous recombination (marker gene deletion) to generate MVA-N. MVA-SARS-2-ST/N (MVA-ST/N): The site of deletion III was targeted by inserting the gene sequence encoding the stabilized S-protein (ST) of SARS-CoV-2 under transcriptional control of the vaccinia virus promoter PmH5. Repetitive sequences served to remove the mCherry marker gene by intragenomic homologous recombination (marker gene deletion) to generate MVA-SARS-2-ST (MVA-ST) [10]. The N-protein of SARS-CoV-2 was inserted into MVA-ST as described for MVA-N to generate MVA-ST/N. (**B**) Multiple-step growth analysis of recombinant MVA-N and MVA-ST/N. CEF or DF-1 cells as well as human HaCat cells were infected at a multiplicity of infection (MOI) of 0.1 with MVA, MVA-N, or MVA-ST/N and samples were collected at the indicated time points. Samples were titrated on CEF monolayers, and plaque-forming units (PFUs) were determined. Differences between the groups were analyzed, determining the area under the curve (AUC) prior to analysis by Kruskal–Wallis Test. (**C**,**D**) Synthesis of the SARS-2-ST and SARS-2-N protein by MVA-N and MVA-ST/N. (**C**) Western blot analysis of SARS-2-S2 and SARS-2-N in lysates of MVA-ST/N-infected cells indicating synthesis of S- and N-protein and Western blot analysis of SARS-2-N in lysates of MVA-N-infected cells indicating synthesis of N-protein. DF-1 cells were infected with an MOI of 5 with mock control, MVA, MVA-N, or MVA-ST/N and collected 0, 4, 8, 12, 24, and 48 h post infection (hpi). Polypeptides in cell lysates were detected using a monoclonal antibody against SARS-2-S2 and SARS-2-N. (**D**) Western blot analysis of SARS-2-S2 and SARS-2-N in lysates of MVA-N- and MVA-ST/N-infected cells indicating synthesis of S- and N-protein. Analysis of Hsp90 served as positive control. DF-1 cells were infected with an MOI of 5 with mock control, MVA, MVA-N, or MVA-ST/N and collected 24 h post infection (hpi). Polypeptides in cell lysates were detected using a monoclonal antibody against SARS-2-S2 and SARS-2-N. (**E**) Double immunofluorescent staining for SARS-2-N and SARS-2-S2 in MVA-N- and MVA-ST/N-infected Vero cells (MOI = 0.1; 17 hpi). Non-infected Vero cells (mock) and cells infected with non-recombinant MVA (MVA) served as controls. Permeabilized infected cells were probed with a monoclonal antibody directed against SARS-2-N or the S2-domain of SARS-CoV-2. Polyclonal goat anti-mouse antibody or polyclonal goat anti-rabbit antibody served to visualize red and green fluorescence for N-specific (green) and S-specific (red) fluorescence staining. Cell nuclei were counterstained with DAPI (blue). The yellowish color in the merge image indicates the overlapping of the red (S-protein) and green (N-protein) colors. Scale bar: 50 μm.

**Figure 2 viruses-16-00417-f002:**
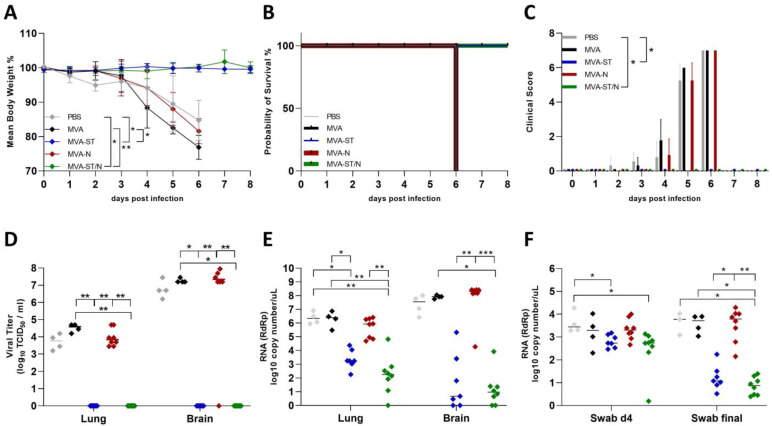
Clinical scoring and viral load after SARS-CoV-2 infection of vaccinated mice. K18-hACE2 mice were intranasally infected with SARS-CoV-2 28 days after intramuscular single vaccination with MVA-ST (*n* = 7), MVA-N (*n* = 8), or MVA-ST/N (*n* = 8) with a dose of 1 × 10^8^ PFU. Empty MVA vector control (*n* = 4) or PBS (mock-vaccinated animals, *n* = 4) were used as controls. Body weight changes, clinical scores, viral loads, and immunogenicity were determined. (**A**) Body weight change, (**B**) survival, and (**C**) clinical symptoms were monitored daily after challenge infection. (**D**) Lungs and brains were taken at the time point of death and analyzed for the amounts of infectious SARS-CoV-2 using TCID_50_ assay. (**E**) Lungs and brains were analyzed for viral RNA (RDRP) of SARS-CoV-2 via RT-qPCR. (**F**) At 4 days post infection and on the day of death, oropharyngeal swabs were taken and analyzed for viral RNA (RDRP) of SARS-CoV-2. Differences between the groups were analyzed, determining the area under the curve (AUC) (**A**) prior to analysis by Kruskal–Wallis Test (**A**,**C**–**F**). Asterisks represent statistically significant differences between two groups: * *p* < 0.05, ** *p* < 0.01, *** *p* < 0.001.

**Figure 3 viruses-16-00417-f003:**
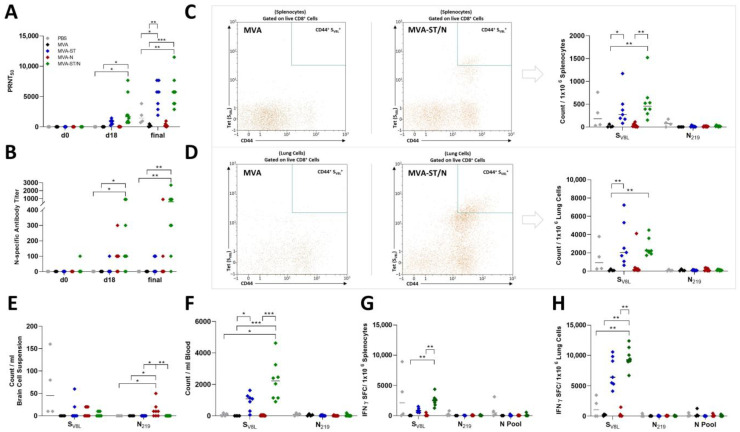
Immune responses of vaccinated mice after infection with SARS-CoV-2. K18-hACE2 mice were challenged with SARS-CoV-2 28 days after intramuscular vaccination with 1 × 10^8^ PFU of MVA-ST (*n* = 7), MVA-N (*n* = 8), MVA-ST/N (*n* = 8), empty MVA (*n* = 4), or PBS (*n* = 4). Serum was taken before (d0), 18 days (d18) after immunization, as well as upon death. Cells were isolated from spleens, lungs, and brains on the day of death. Sera were analyzed for (**A**) neutralizing antibodies with PRNT_50_ assay as well as for (**B**) N-specific antibodies with ELISA. (**C**–**F**) Splenocytes (**C**), lung cells (**D**), brain cells (**E**), and whole blood samples (**F**) were analyzed for specific T cells against the immunodominant peptide of the S-protein (S_V8L_) and the N-protein (N_219_) of SARS-CoV-2 using flow cytometry. (**G**,**H**) Splenocytes (**G**) and lung cells (**H**) were additionally analyzed after stimulation with S_V8L_, N_219_, or N-peptide pool (N Pool) for IFN-γ spot-forming cells (SFCs) measured by ELISPOT assay. Differences between the groups were analyzed by Kruskal–Wallis Test. Asterisks represent statistically significant differences between two groups: * *p* < 0.05, ** *p* < 0.01, *** *p* < 0.001.

**Figure 4 viruses-16-00417-f004:**
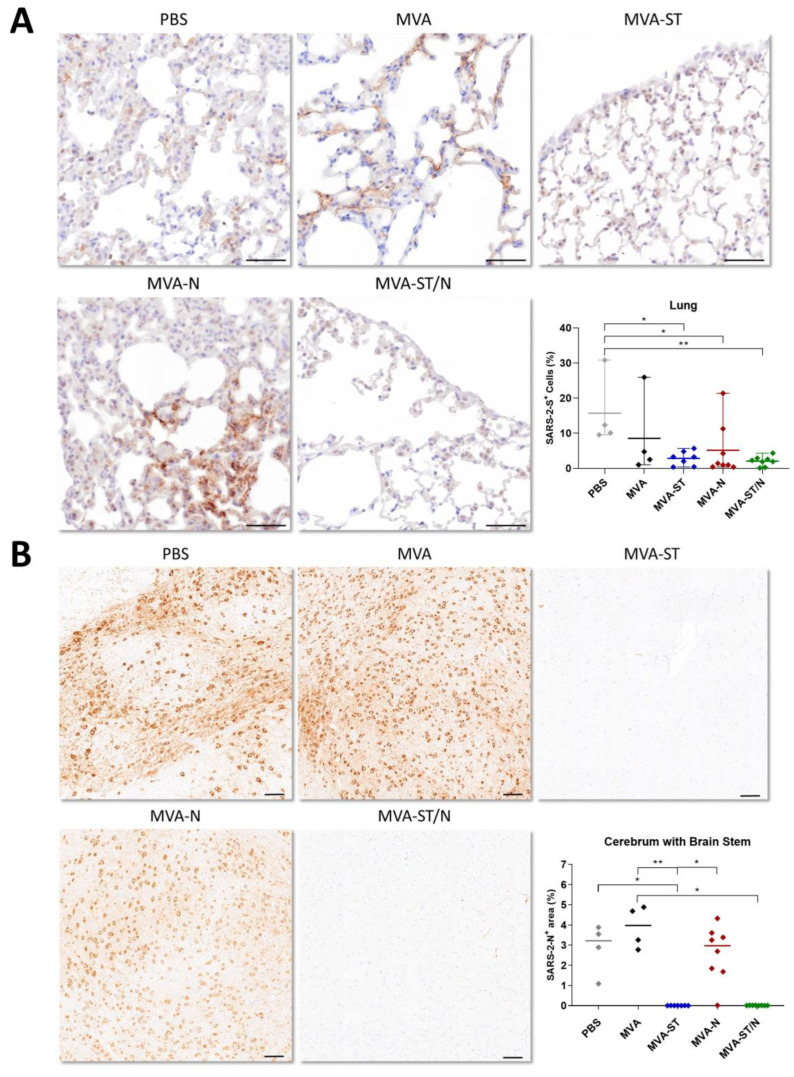
Immunohistochemistry of lung and brain tissue after single vaccination. Lungs and brains were sampled 6–8 days post challenge infection with SARS-CoV-2 and stained for analysis of SARS-CoV-2-antigen-positive cells. (**A**) Staining of lung sections for SARS-2-S antigen. While the overall signal within the vaccination groups remained low in comparison to PBS-treated animals, high amounts of immunopositive cells could be observed multifocally in an individual animal of PBS, MVA, and MVA-N groups. High magnification pictures of stained lung tissue, scale bars: 50 μm. (**B**) Staining of brain sections for SARS-2-N+ showing numerous positive neurons spreading throughout the thalamus of animals from PBS, MVA, and MVA-N groups, in contrast to those immunized with MVA-ST or MVA-ST/N. Accordingly, groups treated with MVA-ST and MVA-ST/N show a significant decrease in positive-SARS-2-N+ area% in comparison to the PBS, MVA, and MVA-N groups in the cerebrum and brain stem. High magnification pictures of stained brain tissue, scale bars: 100 μm. Differences between the groups were analyzed by Kruskal–Wallis Test. Asterisks represent statistically significant differences between two groups: * *p* < 0.05, ** *p* < 0.01.

**Figure 5 viruses-16-00417-f005:**
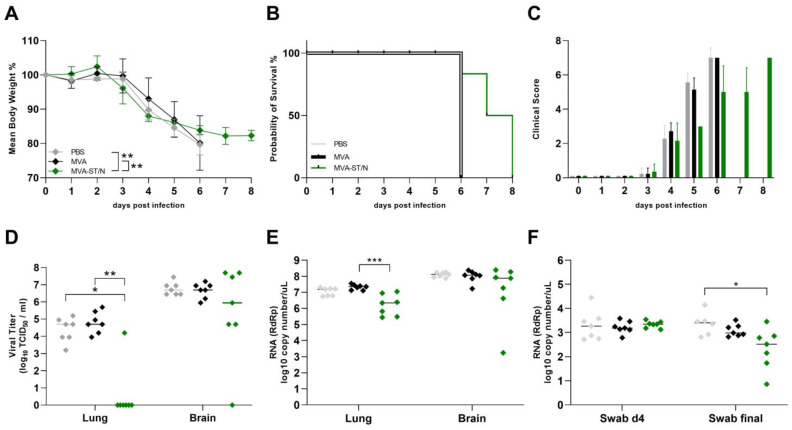
Clinical scoring and viral load after SARS-CoV-2 infection of emergency-vaccinated mice. K18-hACE2 mice were challenged with SARS-CoV-2 two days after intramuscular vaccination with MVA-ST/N (10^8^ PFU, *n* = 7), empty MVA (*n* = 7), or PBS (mock-vaccinated animals, *n* = 7). Body weight changes, clinical scores, viral loads, and immunogenicity were determined. (**A**) Body weight change, (**B**) survival, and (**C**) clinical symptoms were monitored daily after challenge infection. (**D**) Lungs and brains were taken at the time point of death and analyzed for the amounts of infectious SARS-CoV-2 using TCID_50_ assay. (**E**) Lungs and brains were analyzed for viral RNA (RDRP) of SARS-CoV-2 via RT-qPCR. (**F**) At 4 days post infection and on the day of death, oropharyngeal swabs were taken and analyzed for viral RNA (RDRP) of SARS-CoV-2. Differences between the groups were analyzed, determining the area under the curve (AUC) (**A**) prior to analysis by One-way ANOVA (**A**,**C**) or Kruskal–Wallis Test (**D**–**F**). Asterisks represent statistically significant differences between two groups: * *p* < 0.05, ** *p* < 0.01, *** *p* < 0.001.

**Figure 6 viruses-16-00417-f006:**
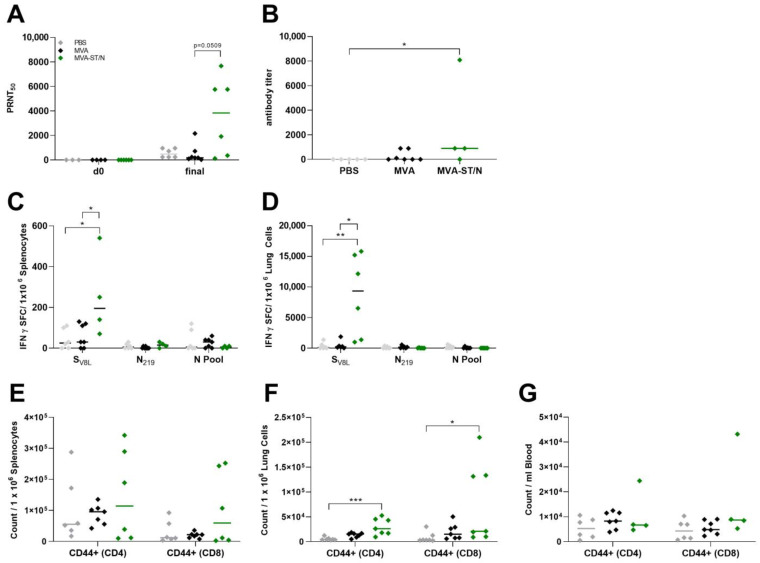
Immune responses of emergency-vaccinated mice after infection with SARS-CoV-2. K18-hACE2 mice were challenged with SARS-CoV-2 two days after intramuscular vaccination with MVA-ST/N (10^8^ PFU, *n* = 7), empty MVA (*n* = 7), or PBS (mock-vaccinated animals, *n* = 7). Serum was taken right before immunization (d0) and after death (final) and cells were isolated from blood, spleens, and lungs after death. Sera were analyzed for (**A**) neutralizing antibodies with PRNT_50_ assay as well as for (**B**) N-specific antibodies in final serum with ELISA. (**C**,**D**) Splenocytes and lung cells were analyzed after stimulation with the immunodominant peptide of the S-protein (S_V8L_), the N-protein (N_219_), or N peptide pool (N Pool) of SARS-CoV-2 for IFN-γ spot-forming cells (SFCs) measured by ELISPOT assay. (**E**–**G**) Splenocytes, lung cells, and whole blood samples were analyzed for activated T cells using flow cytometry. Differences between the groups were analyzed by Kruskal–Wallis Test. Asterisks represent statistically significant differences between two groups: * *p* < 0.05, ** *p* < 0.01, *** *p* < 0.001.

**Figure 7 viruses-16-00417-f007:**
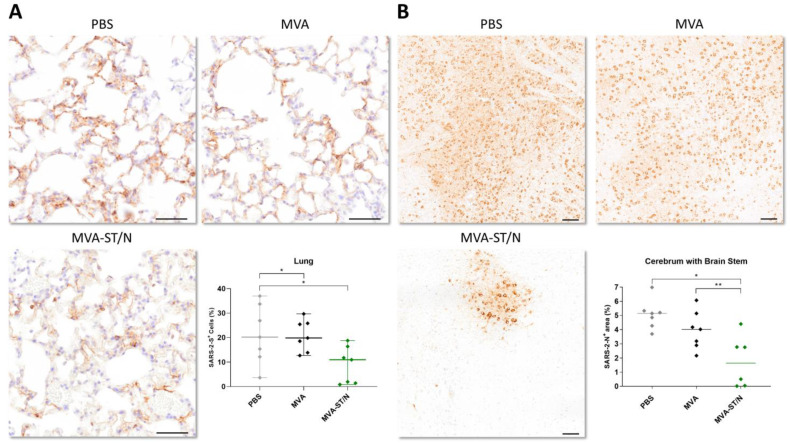
Immunohistochemistry of lung and brain tissue after emergency vaccination. Lungs and brains were sampled 6–8 days post challenge infection with SARS-CoV-2 and stained for analysis of SARS-CoV-2-positive cells. (**A**) Staining of lung sections for SARS-2-S+ cells. Animals receiving PBS and MVA treatment showed high amounts of SARS-2-S+ cells within the alveolar compartment, while a significant reduction in antigen-positive cells was observed in animals that had received the MVA-ST/N vaccine. High magnification pictures of stained lung tissue, scale bars: 50 μm. (**B**) Staining of brain sections for SARS-2-N+ cells. Most of the control-vaccinated animals show coalescing to diffuse distribution of positive neurons, in contrast to the immunized group with MVA-ST/N, which presented only multiple foci of immunoreactive cells. Accordingly, the group treated with MVA-ST/N shows a significant decrease in positive-SARS-2-N area% in comparison to the PBS and MVA groups in the cerebrum and brain stem. High magnification pictures of stained brain tissue, scale bars: 100 μm. Differences between the groups were analyzed by One-way ANOVA. Asterisks represent statistically significant differences between two groups: * *p* < 0.05, ** *p* < 0.01.

## Data Availability

The datasets generated and/or analyzed during the current study are available from the corresponding authors upon reasonable request.

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
