# Peer review of "Single MVA-SARS-2-ST/N Vaccination Rapidly Protects K18-hACE2 Mice against a Lethal SARS-CoV-2 Challenge Infection"

_viruses, 2024, doi:10.3390/v16030417_

Round 1

Reviewer 1 Report

Comments and Suggestions for Authors

Clever et al. address the need for effective vaccines in response to the emergence of new SARS-CoV-2 variants. The authors designed different vaccines based on a recombinant Modified Vaccinia Ankara (MVA) vaccine expressing either the prefusion-stabilized spike protein (ST) or the nucleoprotein (N) of SARS-CoV-2 or both applied by a combinatory MVA vector approach. The paper describes besides the construction and characterization of the vectors two immunization studies in mice using single systemic vaccination approach, either 28 days before infection or as an emergency vaccination at 2 days before infection. A comprehensive analysis of the humoral and cellular immune response, as well as protective efficacy against SARS-CoV-2 challenge infection were performed.

Major points:

-          The main motivation behind exploring the combinatory vaccine approach was to confer protection against emerging SARS-CoV-2 VOCs with mutated S proteins. Consequently, the mouse challenge studies should have been designed to involve infection using at minimum one of these VOC. As an alternative, neutralization assays using sera from vaccinated mice against SARS-CoV-2 VOCs to substantiate the vaccine's potential efficacy against these VOCs would be beneficial, as shown by the authors before (reference 10).

-          The comparison of vaccines expressing either ST or N protein alone to the vaccine expressing both proteins did not clearly demonstrate a benefit of the MVA-ST/N protein in terms of protection when compared to the MVA-ST vaccine. Only slight improvement in some immunogenicity parameters was observed, except humoral immune responses against SARS-CoV-2 N-protein. is different to Therefore, the rational for the usage of MVA-ST/N for the second immunization trial is not clearly explained. The authors are encouraged to state this more clearly also in the discussion section.

-          The usage of different vaccine candidates is highly warranted. In that line, the direct comparison of MVA-N and MVA-N/ST reveals somehow, that the combination with ST and N gene in the context of MVA stimulates the immune response towards the N antigen and S antigen (Figure 3). Although these effects seem to be not statistically significant the differences are strikingly shown for humoral and cellular immune responses. Would this also be the case by simply mixing both vaccine candidates MVA-N and MVA-ST without changing the vaccine vector titer? The authors are kindly asked to add this aspect into their discussion as well.  

-          In the abstract, the authors claim that the single emergency vaccination with MVA-ST/N resulted in significant titers of neutralizing antibodies. However, when looking at Fig. 6A, the increase in titer is not statistically significant, observed only in three out of seven mice, and solely after the challenge infection “after a boost by the challenge infection”. Also, in line 937-938, the authors mention a robust activation of virus-neutralizing antibodies and N-antibody titer after the emergency vaccination, despite the fact that the anti-N antibodies were elevated only in one mouse. This should be reformulated in the abstract and in the whole text and stated more cautiously.

-          The analysis of T-cell-specific immune responses in immunogenicity studies is typically performed on vaccinated mice that have not undergone challenge infection to distinguish between the vaccination and other immune response reactivations. In this study, the measured cellular immune responses are not solely a result of vaccination but also involves a boost through the SARS-CoV-2 challenge infection. This distinction should be included in data interpretation in the results section as well as in the discussion.

Minor points:

-          The presentation of Material and Method section should be modified. Please add sections with corresponding headlines.

-          Since emergency vaccination was shown before to be protective 4 days before challenge the authors should explain why they used two days before challenge.

-          Figures 5 and 6: Is it a mistake, or have the author used here to compare the MVA vector vaccines MVA-S/N instead of MVA-ST/N. Please correct the graphs.

-          Please clarify the reason behind the varying sample sizes used in the first immunization study (n = 7, 8, 4)?

-          Line 221: provide a more detailed description of the scoring system used to evaluate disease severity in mice and elaborate on the criteria for euthanasia. Probably insert one line into the graphs (Figures 2C and 5C) for the defined criteria of euthanasia.

-          Figure 3 A, B, C and D. Unclear what details are presented by the graphs in the upper part and the lower part of each of the graphs. Please either put it in one graph or give it its own letter. Why the sera were not tested for binding to the S protein, but only to the N-protein?

-          Gray color on 3G and H too fade. Harmonize grey color to the other graphs.

-          Figures 3B and 6B: The mouse sera were tested for determination of antibody titer against N protein. Please add antibody titer against S protein or discuss?

-          According to Fig. 5C, the termination score for mice in the control group (PBS and MVA) is at 7 points. However, when looking at the last day of the study (day 8), all surviving mice from the MVA-S/N vaccinated group reached also the score of 7. If these mice met euthanasia criteria on the last day of the study, they should not be categorized as survivors, and the survival curve has to be adjusted accordingly.

-          Figure 1B is too small. The reader will be unable to read the descriptions on the graph.

-          Fig. 1E: Include the scale on the microscopic pictures.

-          Fig 2B, 5B: For better visibility the authors may use an off-set of the survival lines.

-          Fig. 2C, 5C: The authors are asked to use lines of mean values and standard deviation for the clinical scores.

-          Figures 4 and7. How are the measurements for the graphs determined? Please describe more detailed how the sections of the immune histochemistry pictures were used for the read-out parameters shown in the graphs. In addition, as the graphs on 4B and 7B have a header please harmonize as well for 4A and 7A.

-          Fig. 6B: Specify whether the antibody titer is measured on day 0 or at the final day. Please add in the figure legend that day 0 is the time point of immunization, not before death.

-          Fig. 6: in several figures, the displayed data points are less than the given size of the group (n=7). For example, in Figure 6B, the MVA-S/N group shows only 3 data points, and the PBS group shows only 4 data points. Please correct and/or explain.

-          Line 702: the statement "MVA-ST/N resulted in the absence of infectious SARS-CoV-2" should include “except for one mouse” for accuracy.

-          The authors might also include as supplement figure the gating strategy for the FACS-analysis of the cytotoxic T-cells.

-          Discussion section: Line 813: The expression of SARS-CoV-2 S-protein instead of ST-protein.

-          Line 819: “vaccine candidate” instead of “candidate vaccine”

-          Line 832-836: The authors include references of previously used MVA vaccines with combination of S and N. Please explain the difference to the presented vaccines and the differences to the published data.

In summary, I recommend the manuscript for publication after the proposed revisions, as it will contribute to the ongoing efforts in the field of SARS-CoV-2 vaccines.

Author Response

Point-by-Point response to the reviewers`comments (viruses-2889978)

We thank the reviewers and the editors for their insightful observations and comments, which have all been answered and enabled us to resubmit an improved manuscript. Below we listed all reviewers’ points in black and our answers are in blue. Within the manuscript, edits in response to the reviewers’ comments are highlighted in yellow.

Reviewer 1:

Clever et al. address the need for effective vaccines in response to the emergence of new SARS-CoV-2 variants. The authors designed different vaccines based on a recombinant Modified Vaccinia Ankara (MVA) vaccine expressing either the prefusion-stabilized spike protein (ST) or the nucleoprotein (N) of SARS-CoV-2 or both applied by a combinatory MVA vector approach. The paper describes besides the construction and characterization of the vectors two immunization studies in mice using single systemic vaccination approach, either 28 days before infection or as an emergency vaccination at 2 days before infection. A comprehensive analysis of the humoral and cellular immune response, as well as protective efficacy against SARS-CoV-2 challenge infection were performed.

We greatly appreciate the reviewer’s positive response.

Major points:

-          The main motivation behind exploring the combinatory vaccine approach was to confer protection against emerging SARS-CoV-2 VOCs with mutated S proteins. Consequently, the mouse challenge studies should have been designed to involve infection using at minimum one of these VOC. As an alternative, neutralization assays using sera from vaccinated mice against SARS-CoV-2 VOCs to substantiate the vaccine's potential efficacy against these VOCs would be beneficial, as shown by the authors before (reference 10).

We fully agree with the reviewer. The initial motivation behind exploring the combinatory approach was to broaden the immune response against different SARS-CoV-2 variants of concern. Since this has been confirmed for previous studies, which also confirmed an improved immunogenicity including robust T cell responses, we aimed to evaluate whether the improved immunogenicity was also able to rapidly protect against SARS-CoV-2 challenge infection. The reason behind this hypothesis is that in an orthopoxvirus background, we identified an essential role of T cells to confer rapid protection. For this, we aimed to evaluate a more rapid protective capacity induced by MVA-SARS-2-ST/N. To contain a baseline for direct comparison with previous studies, we started with the original SARS-CoV-2 Wuhan isolate. Due to the low amount of sera from the mice in these studies, we were not able to perform any neutralisation assays with the variants. The evaluation of the cross-protective efficacy of our vaccine also against other relevant variants will be very interesting for future studies. To make this point clear, we included a section within the discussion: Line 1014-10016: „In that context, it will be of significant importance to also evaluate the rapid protective capacity against different variants of SARS-CoV-2 also including VOCs“.

-          The comparison of vaccines expressing either ST or N protein alone to the vaccine expressing both proteins did not clearly demonstrate a benefit of the MVA-ST/N protein in terms of protection when compared to the MVA-ST vaccine. Only slight improvement in some immunogenicity parameters was observed, except humoral immune responses against SARS-CoV-2 N-protein. is different to Therefore, the rational for the usage of MVA-ST/N for the second immunization trial is not clearly explained. The authors are encouraged to state this more clearly also in the discussion section.

We fully agree with the reviewers points, we did not detect an exceptional improved outcome of our combinatory vaccine compared to the MVA-SARS-2-ST vaccine. Nevertheless, in the direct comparison, we observed the tendency to induce an improved immunogenicity after vacination with regard to antibodies and T cells and also a tendency to better reduction of the viral load in upper and lower respiratory tract. We included in the discussion at line 967-970: „Based on the slightly improved effects for immunogenicity and efficacy seen in the MVA-SARS-2-ST/N vaccine in the single vaccination approach, we aimed to evaluate the MVA-SARS-2-ST/N vaccination in a shortened interval.“

-          The usage of different vaccine candidates is highly warranted. In that line, the direct comparison of MVA-N and MVA-N/ST reveals somehow, that the combination with ST and N gene in the context of MVA stimulates the immune response towards the N antigen and S antigen (Figure 3). Although these effects seem to be not statistically significant the differences are strikingly shown for humoral and cellular immune responses. Would this also be the case by simply mixing both vaccine candidates MVA-N and MVA-ST without changing the vaccine vector titer? The authors are kindly asked to add this aspect into their discussion as well. 

This is a valid question and we fully understand the reviewer’s point. We hypothesize that we would also see the effect when mixing both vaccine candidates without changing the vaccine vector titer. However, since the volume would be double to result in the same vaccine dosage, it is not possible to evaluate this effect in mice since we are restricted due to animal protection law in Germany. To highlight this aspect, we included this information now in the discussion: Line 1019-1024: “ In summary, these data further highlight the promising potential of a multivalent vaccine co-expressing S- and N-protein as confirmed in a lethal mouse model for COVID-19. Using a multivalent MVA-vaccine co-expressing the S-and N-protein for efficacy testing in mice, we also overcome the difficulties of simply mixing both vaccine candidates MVA-N and MVA-ST, which is not possible due the limiting volumes for vaccination in mice.“.

-          In the abstract, the authors claim that the single emergency vaccination with MVA-ST/N resulted in significant titers of neutralizing antibodies. However, when looking at Fig. 6A, the increase in titer is not statistically significant, observed only in three out of seven mice, and solely after the challenge infection “after a boost by the challenge infection”. Also, in line 937-938, the authors mention a robust activation of virus-neutralizing antibodies and N-antibody titer after the emergency vaccination, despite the fact that the anti-N antibodies were elevated only in one mouse. This should be reformulated in the abstract and in the whole text and stated more cautiously.

This was a good suggestion. We fully agree with the reviewer that the differences are not statistically significant and are limited to a small number of mice. To make this more clear, we reformulated the description of humoral immune responses throughout the manuscript and the abstract.

-          The analysis of T-cell-specific immune responses in immunogenicity studies is typically performed on vaccinated mice that have not undergone challenge infection to distinguish between the vaccination and other immune response reactivations. In this study, the measured cellular immune responses are not solely a result of vaccination but also involves a boost through the SARS-CoV-2 challenge infection. This distinction should be included in data interpretation in the results section as well as in the discussion.

We agree with the reviewer. More detailed studies on the analysis of cellular immune responses also without challenge infection signifcantly contributes to the characterization and interpretation of the different candidate vaccines. To make this more clear, we highlighted in the results section that T cell responses have been analyzed after challenge infection (Line 622-623 and 768-769). Moreover, we also included a section in the discussion evaluating on the impact of the SARS-CoV-2 challenge infection as an additional boost vaccination. Line 997-1000: „Moreover, future studies evaluating the early activation and kinetics of T cell responses without the SARS-CoV-2 challenge infection which might serve as a boost effect should be characterized.“ We also included a section to demonstrate the need for future clinical studies characterizing the levels and kinetics of T cells without challenge infection: Line 948-953 „Of note, to evaluate the role of T cells induced by single vaccination with the different MVA-SARS-2 candidate vaccines in more detail and define possible immunological correlates of protection, future studies characterizing the cellular immune responses without challenge infection to differentiate between the vaccination and other immune response reactivations, are warranted.“

Minor points:

-          The presentation of Material and Method section should be modified. Please add sections with corresponding headlines.

This is a very appropriate suggestion and we now have modified this appropriately.

-          Since emergency vaccination was shown before to be protective 4 days before challenge the authors should explain why they used two days before challenge.

Based on our previous studies using MVA as an emergency vaccine to rapidly protect mice against a lethal mousepox challenge infection 2 days after the initial vaccination, we aimed to evaluate the emergency protective capacity of MVA-SARS-2-ST/N also 2 days after initial vaccination. These studies are mentioned within the introduction and discussion (see Ref 60 and 61). Line 879-881: „In previous studies, we already confirmed the rapid protective capacity of a single MVA vaccination against lethal orthopoxvirus challenge infection.“

-          Figures 5 and 6: Is it a mistake, or have the author used here to compare the MVA vector vaccines MVA-S/N instead of MVA-ST/N. Please correct the graphs.

We are sorry for this mistake, we changed the nomenclature of MVA-S/N to MVA-ST/N.

-          Please clarify the reason behind the varying sample sizes used in the first immunization study (n = 7, 8, 4)?

Based on the different effects exspected to see in the vaccination groups vs. the control groups, the statistical computation resulted in varying numbers of animals per group. This is in accordance with the German animal protection law and the 3-R-strategy to reduce the number of animals without loosing statistical significance. The experiment was planned and statistically designed with group sizes of n=8 for the vaccination groups (MVA-ST, MVA-N, MVA-ST/N) and group sizes of n=4 for the control groups (PBS, MVA). Unfortunately, one mouse of the MVA-ST group died unexpectedly during anesthesia.

-          Line 221: provide a more detailed description of the scoring system used to evaluate disease severity in mice and elaborate on the criteria for euthanasia. Probably insert one line into the graphs (Figures 2C and 5C) for the defined criteria of euthanasia.

This is a very appropriate suggestion and we included more detailed information for the scoring system in the methods. Line 236-243: „Symptoms were assigned to the following categories: cardiovascular system, fur / skin condition, respiratory tract, social behavior / general condition / locomotion and neuro-logical abnormalities. Respiratory signs were additionally divided into upper and lower respiratory tract with focus on nasal discharge and tachypnoea respectively. Clinical symptoms were assigned for clinical scores using the clinical score sheet and further combined as a total clinical score for each mouse. Exceeding weight loss from the initial body weight over 20% as well as a defined cumulative clinical score were defined as ex-perimental endpoint.“

-          Figure 3 A, B, C and D. Unclear what details are presented by the graphs in the upper part and the lower part of each of the graphs. Please either put it in one graph or give it its own letter. Why the sera were not tested for binding to the S protein, but only to the N-protein?

This is a very appropriate suggestion. We merged the plots of Figure 3A and 3B and framed the flow cytometry plots and figures tu ensure better affiliation.

Due to the very limited amount of sera obtained from mice, we had to prioritize the assays for the characterization of specific antibodies. Since the titers of neutralizing antibodies are still considered to be a gold standard for vaccine evaluation, we decided to perform neutralizing assays. Since the N-protein does not activate neutralizing antibodies, we used here the characterization of binding antibodies.

-          Gray color on 3G and H too fade. Harmonize grey color to the other graphs.

This was a good suggestion and we harmonized the grey color appropriately.

-          Figures 3B and 6B: The mouse sera were tested for determination of antibody titer against N protein. Please add antibody titer against S protein or discuss?

This point mimics a point raised above. Since the available volumes of sera were not sufficient for all assays, we had to limit the analysis to the neutralizing antibodies directed against SARS-CoV-2. We have included this limitation in the discussion. Lines: 915-916 „Here, it will of interest for future studies to also characterize the titers of S-binding an-tibodies induced by the MVA-SARS-2-ST/N vaccine.“

-          According to Fig. 5C, the termination score for mice in the control group (PBS and MVA) is at 7 points. However, when looking at the last day of the study (day 8), all surviving mice from the MVA-S/N vaccinated group reached also the score of 7. If these mice met euthanasia criteria on the last day of the study, they should not be categorized as survivors, and the survival curve has to be adjusted accordingly.

This is a very appropriate suggestion. We changed the figure appropriately.

-          Figure 1B is too small. The reader will be unable to read the descriptions on the graph.

This is a very appropriate suggestion. We changed the figure appropriately.

-          Fig. 1E: Include the scale on the microscopic pictures.

This is a very appropriate suggestion. We changed the figure appropriately.

-          Fig 2B, 5B: For better visibility the authors may use an off-set of the survival lines.

This is a very appropriate suggestion. We changed the figure appropriately.

-          Fig. 2C, 5C: The authors are asked to use lines of mean values and standard deviation for the clinical scores.

This is a very appropriate suggestion. We changed the figure appropriately.

-          Figures 4 and7. How are the measurements for the graphs determined? Please describe more detailed how the sections of the immune histochemistry pictures were used for the read-out parameters shown in the graphs. In addition, as the graphs on 4B and 7B have a header please harmonize as well for 4A and 7A.

This is a very appropriate suggestion. We included additional information in the methods (Lines 403-426) and changed the figure appropriately.

-          Fig. 6B: Specify whether the antibody titer is measured on day 0 or at the final day. Please add in the figure legend that day 0 is the time point of immunization, not before death.

This is a very appropriate suggestion. We changed the figure appropriately.

-          Fig. 6: in several figures, the displayed data points are less than the given size of the group (n=7). For example, in Figure 6B, the MVA-S/N group shows only 3 data points, and the PBS group shows only 4 data points. Please correct and/or explain.

This is a valid question and we fully understand the reviewer’s point. Unfortunately, a few sera could not be analyzed due to poor quality. Nevertheless, we were able to analyze the sera of the other animals well and fortunately, we still saw these tendencies of differences between the groups.

-          Line 702: the statement "MVA-ST/N resulted in the absence of infectious SARS-CoV-2" should include “except for one mouse” for accuracy.

This is a very appropriate suggestion. We changed the statement appropriately.

-          The authors might also include as supplement figure the gating strategy for the FACS-analysis of the cytotoxic T-cells.

This is a very appropriate suggestion. We included the gating strategy for the FACS analysis of the cytotxic T cells in the supplementary figure (Figure S2).

-          Discussion section: Line 813: The expression of SARS-CoV-2 S-protein instead of ST-protein.

This is a very appropriate suggestion. We changed the sentence appropriately.

-          Line 819: “vaccine candidate” instead of “candidate vaccine”

This is a very appropriate suggestion. We changed the sentence appropriately.

-          Line 832-836: The authors include references of previously used MVA vaccines with combination of S and N. Please explain the difference to the presented vaccines and the differences to the published data.

These references in principle all focus on the evaluation of the immunogenictiy and efficacy of an MVA vector virus expressing the S- and N-protein in the same vaccine vector. These studies have been performed using different MVA backbone viruses for construction of the recombinant MVA viruses and also different backbone sequences encoding for the S- and the N-proteins have been used. Moreover, different animal models have been used for the characterizatin of the in vivo safety, immunogencitiy and efficacy against different SARS-CoV-2 viruses also including VOCs. This might contribute to the differences seen in the different MVA-bases vaccines. In doing so, Chiuppesi et al (Ref. 23) used a native S-protein within the candidate vaccine COH04S1, which is based on the sMVA platform using the BAC cloning for recombinant virus vaccine formation. They characterized the MVA-candidate vaccines in the the Syrian hamster and in Non-Human Primates (NHP). For challenge, they used the SARS-CoV-2 USA-WA1/2020 strain. Routhu et al (Ref. 28) used a stabilized version of the S-protein within the vaccine, but used other deletion sites of MVA as insert destination for the target seuquences, compared to our vaccines. Routhu et al tested the multivalent vaccine in NHP and performed challenge infection with the SARS-CoV-2 delta variant.

In summary, I recommend the manuscript for publication after the proposed revisions, as it will contribute to the ongoing efforts in the field of SARS-CoV-2 vaccines.

Reviewer 2 Report

Comments and Suggestions for Authors

In this manuscript, Clever et al., investigated the MVA-based vaccine efficacy against SARS-CoV-2. They immunized mice with vaccine which encoded SARS-CoV-2 S and N and found that the immunization (especially S and S+N) induced strong SARS-CoV-2 specific CD8 responses and decreased virus in mice. Overall, the manuscript and experiments were well organized and written. I have a few minor comments on this manuscript.

1.     Figure 3C.and D: The authors should add control panel to see the expanding Tet+ CD8 T cells. 

2.     Figure 1E: Please add the scale bar.

3.     Some words in the Figure 1A right and 1B are too small to read. It’s much better for reader to use bigger font.

4.     In the survival data, the authors showed that the vaccine protected/treated mice from infection (at least 8-dpi). Do the authors think MVA-based vaccine induced long-term protection? Please add discussion.

Comments on the Quality of English Language

Minor editing and spell check are required.

Author Response

Reviewer 2:

In this manuscript, Clever et al., investigated the MVA-based vaccine efficacy against SARS-CoV-2. They immunized mice with vaccine which encoded SARS-CoV-2 S and N and found that the immunization (especially S and S+N) induced strong SARS-CoV-2 specific CD8 responses and decreased virus in mice. Overall, the manuscript and experiments were well organized and written. I have a few minor comments on this manuscript.

We greatly appreciate the reviewer’s positive response.

  1. Figure 3C.and D: The authors should add control panel to see the expanding Tet+ CD8 T cells.

This is a very appropriate suggestion and we changed Figure 3C and 3D appropriately.

  1. Figure 1E: Please add the scale bar.

This is a very appropriate suggestion and we included the scale bar.

  1. Some words in the Figure 1A right and 1B are too small to read. It’s much better for reader to use bigger font.

This was a good suggestion. We used a bigger font for Figure 1A and Figure 1B.

  1. In the survival data, the authors showed that the vaccine protected/treated mice from infection (at least 8-dpi). Do the authors think MVA-based vaccine induced long-term protection? Please add discussion.

This is a valid question and we fully understand the reviewer’s point. Based on the titers of antibodies induced by the MVA-SARS-2-ST vaccine in the prime vaccination study, we hypothesize that we would also exspect a long-term protection as already established and confirmed in our previous study (Ref. 10). Titers of neutralizing antibodies induced by MVA-SARS-2-ST candidate vaccine after single shot vaccination are already comparable to the titers of neutralizing antibodies measured after prime-boost vaccination (Ref 10). This hypothesis is further supported by the absence of any infectious virus in upper and lower respiratory tract. Moreover, this long-term effect is also supported by additional studies testing the long-term immunogenicity of the MVA-SARS-2-S vaccine [1]. Future studies also evaluating the long-term efficacy of our MVA-SARS-2-ST/N candidate vaccine will be of interest. We included a section in the discussion. Line 1016-1019: „With regard to a long-term protective efficacy, which will also contribute to minimize the risk for the emergence of new VOCs, future studies also evaluating the immunogenicity and efficacy at late immunization schedules are important and interesting.“.
